# The scheduling of adolescence with Netrin-1 and UNC5C

Daniel Hoops[1,2], Robert Kyne[3], Samer Salameh[2,4], Del MacGowan[2,4], Radu Gabriel Avramescu[1,2], Elise Ewing[2,4], Alina Tao He[2,4], Taylor Orsini[2,4], Anais Durand[2,4], Christina Popescu[2,4], Janet Mengyi Zhao[2,4], Kelcie Shatz[5], LiPing Li[5], Quinn Carroll[5], Guofa Liu[6], Matthew J Paul[3,5†], Cecilia Flores[1,2,7,8*†]

[1]Department of Psychiatry, McGill University, Montréal, Canada; [2]Douglas Mental Health University Institute, Montréal, Canada; [3]Neuroscience Program, University at Buffalo, SUNY, United States; [4]Integrated Program in Neuroscience, McGill University, Montreal, Canada; [5]Department of Psychology, University at Buffalo, SUNY, United States; [6]Department of Biological Sciences, University of Toledo, Toledo, United States; [7]Department of Neurology and Neurosurgery, McGill University, Montréal, Canada; [8]Ludmer Centre for Neuroinformatics & Mental Health, McGill University, Montréal, Canada

**\*For correspondence:**
cecilia.flores@mcgill.ca

†Co-senior author

**Abstract** Dopamine axons are the only axons known to grow during adolescence. Here, using rodent models, we examined how two proteins, Netrin-1 and its receptor, UNC5C, guide dopamine axons toward the prefrontal cortex and shape behaviour. We demonstrate in mice (*Mus musculus*) that dopamine axons reach the cortex through a transient gradient of Netrin-1-expressing cells – disrupting this gradient reroutes axons away from their target. Using a seasonal model (Siberian hamsters; *Phodopus sungorus*) we find that mesocortical dopamine development can be regulated by a natural environmental cue (daylength) in a sexually dimorphic manner – delayed in males, but advanced in females. The timings of dopamine axon growth and UNC5C expression are always phase-locked. Adolescence is an ill-defined, transitional period; we pinpoint neurodevelopmental markers underlying this period.

## eLife assessment

This study addresses an **important**, understudied question using approaches that link molecular, circuit, and behavioral changes. The findings that Netrin-1 and UNC5c can guide dopaminergic innervation from the nucleus accumbens to the cortex during adolescence are **solid**. The data showing that the onset of Unc5 expression is sexually dimorphic in mice, and that in Siberian hamsters environmental effects on development are also sexually dimorphic are also **solid**. Reviewers identified significant gaps in evidence for specificity of Netrin-1 expression, which, if filled, would strengthen the evidence for some of the claims. Future work would also benefit from Unc5C knockdown to corroborate the results and investigation of the cause-effect relationship. This paper will be of interest to those interested in neural development, sex differences, and/or dopamine function.

## Introduction

Adolescence is a critical developmental period involving dramatic changes in behaviour and brain anatomy. The prefrontal cortex, the brain region responsible for our most complex cognitive functions, is still establishing connections during this time (*Gogtay et al., 2004*; *Petanjek et al., 2011*;

*Sowell et al., 2004*). The trajectory of prefrontal cortex development in adolescence determines the vulnerability or resilience of individuals to adolescent-onset psychiatric diseases (*Fuhrmann et al., 2015*; *Keshavan et al., 2014*; *Kessler et al., 2007*; *Kessler et al., 2005*; *Lee et al., 2014*). The age at which this adolescent development occurs therefore represents a critical window during which the brain is particularly susceptible to environmental influences. Traditionally, the onset of adolescence is thought to coincide with puberty (*Hollenstein and Lougheed, 2013*). In humans, the age of pubertal onset has been advancing throughout the 19th, 20th, and 21st centuries, and environmental influences, such as nutrition, can pathologically alter the age of puberty (*Wolf and Long, 2016*). However, it remains entirely unknown whether the neural and cognitive maturational processes of adolescence can also be plastic. Here, we examine how the timing of certain adolescent developmental processes are programmed, and whether this timing can be plastic in response to a natural environmental cue, in parallel with pubertal plasticity.

Dopamine innervation to the prefrontal cortex increases substantially across adolescence, and psychopathologies of adolescent origin prominently feature dopamine dysfunction. Evidence continues to emerge that protracted dopamine innervation is a key neural process underlying the cognitive and behavioural changes that characterise adolescence (*Larsen and Luna, 2018*). The mesocorticolimbic dopamine system – which includes the prefrontal cortex – is unique because not only are connections being formed and lost during adolescence, but there is also long-distance displacement of dopamine axons between brain regions. At the onset of adolescence, both mesolimbic and mesocortical dopamine axons innervate the nucleus accumbens in rodents, but the mesocortical axons leave the accumbens and grow toward the prefrontal cortex during adolescence and early adulthood (*Hoops et al., 2018*; *Reynolds et al., 2018*; *Reynolds et al., 2023*). This is the only known case of axons growing from one brain region to another so late during development (*Reynolds and Flores, 2021*).

The prolonged growth trajectory renders mesocortical dopamine axons particularly vulnerable to disruption. Environmental insults during adolescence (e.g. drug abuse) alter the extent and organisation of dopamine innervation in the prefrontal cortex, leading to behavioural and cognitive changes in mice throughout adulthood (*Drzewiecki and Juraska, 2020*; *Hoops and Flores, 2017*; *Reynolds and Flores, 2021*). These changes often involve cognitive control, a prefrontal function that develops in parallel with dopamine innervation to the cortex in adolescence (*Luna et al., 2015*). Disruption of dopamine innervation frequently seems to result in 'immature' cognitive control persisting through adulthood (*Reynolds and Flores, 2021*).

Here, we examine the guidance of growing dopamine axons to the prefrontal cortex, and its timing. The guidance cue molecule Netrin-1, upon interacting with its receptor DCC, determines *which* dopamine axons establish connections in the nucleus accumbens and which ones leave this region to grow to the prefrontal cortex (*Hoops and Flores, 2017*; *Reynolds and Flores, 2021*; *Reynolds et al., 2023*). We hypothesised that the answers to *how* and *when* this extraordinary developmental feat is achieved may also lie in the Netrin-1 signalling system.

## Part 1: Netrin-1 'paves the way' for dopamine axons in adolescence

To identify the route by which dopamine axons grow from the nucleus accumbens to the medial prefrontal cortex, we visualised dopamine axons in the adult mouse forebrain. We observed that dopamine axons medial to the nucleus accumbens occupy a distinct area and are oriented dorsally toward the cortex (*Figure 1A and B*). Individual fibres can be seen crossing the boundary of the nucleus accumbens shell and joining these dorsally oriented axons (*Figure 1C*). We hypothesised that these are the fibres that grow to the prefrontal cortex during adolescence. If this is correct, the number of dopamine axons oriented dorsally toward the medial prefrontal cortex should continue to increase until adulthood. To test this, we used a modified unbiased stereological approach (*Kim et al., 2011*) where axons are counted only if they crossed the upper and lower bounds of a counting probe. We also measured the average width of the area these axons occupy. We found, in both male and female mice, that the density of dopamine axons does not change between adolescence (21 days of age) and adulthood (75 days of age; *Figure 1D*). However, the width of the area that dopamine axons occupy does change, increasing between adolescence and adulthood (*Figure 1E*). These results indicate that the total number of dopamine axons passing through this area increases over adolescence and that dopamine axons grow to the medial prefrontal cortex via this route.

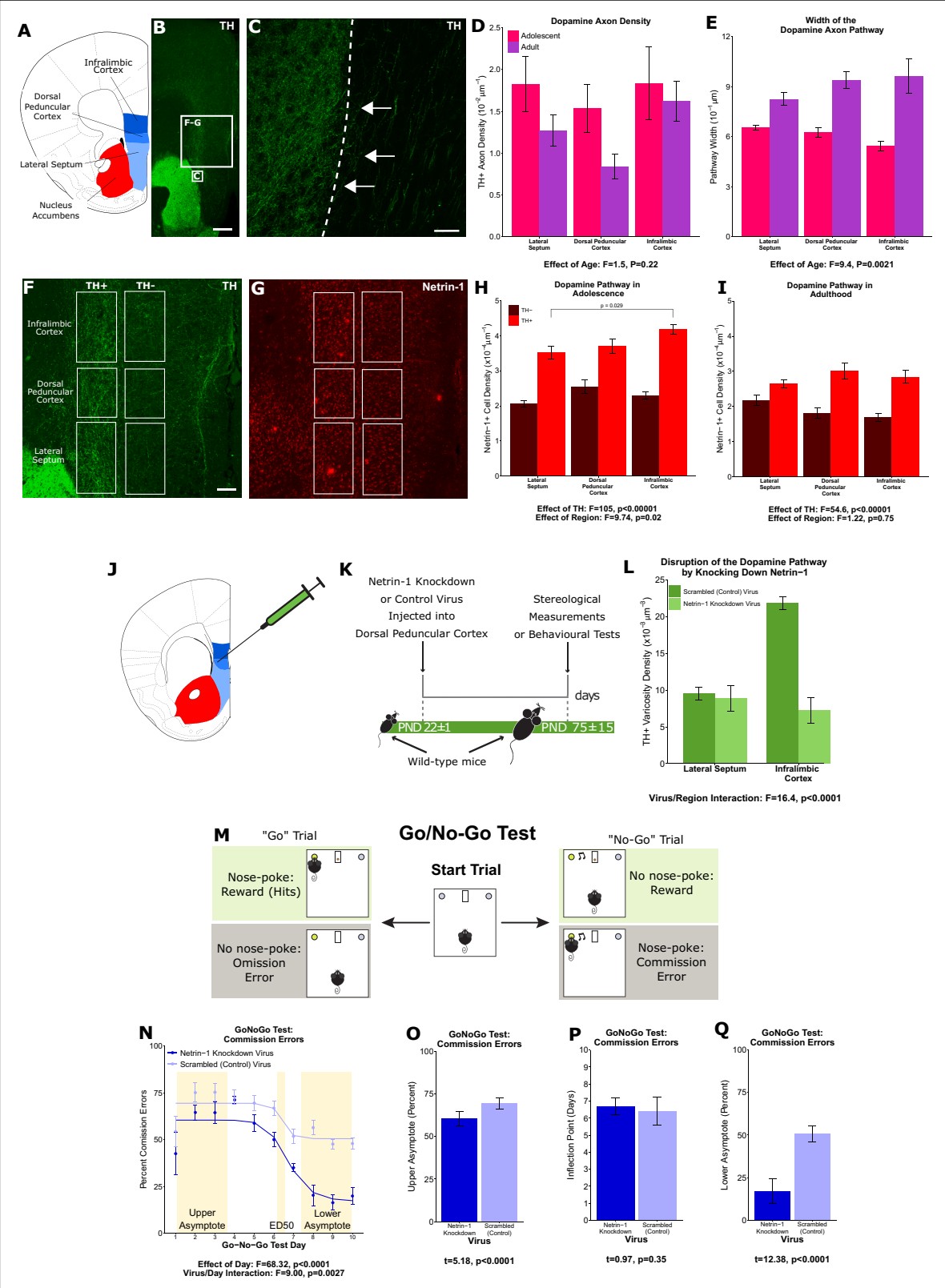

**Figure 1.** A 'pathway' of Netrin-1-expressing cells 'paves the way' for dopamine axons growing from the nucleus accumbens to the medial prefrontal cortex during adolescence. (**A**) The brain regions containing the dopamine fibres passing to the medial prefrontal cortex are highlighted in a line drawing of a coronal mouse brain section derived from *Paxinos and Franklin, 2013*. (**B**) An image of a coronal section through the forebrain of an adult mouse at low magnification (×4). Green fluorescence indicates immunostaining for tyrosine hydroxylase (TH), used here as a marker for dopamine.

*Figure 1 continued on next page*

*Figure 1 continued*

The smaller and larger white squares indicate the regions enlarged in panel C and panels F and G, respectively. Scale bar = 500 µm. (**C**) The nucleus accumbens (left of the dotted line) is densely packed with TH+ axons (in green). Some of these TH+ axons can be observed extending from the nucleus accumbens medially toward TH+ fibres oriented dorsally toward the medial prefrontal cortex (white arrows). Scale bar = 10 µm. (**D**) Modified stereological quantification revealed no significant difference in TH+ axon density between adolescence (21 days of age) and adulthood (75 days of age). Mixed-effects analysis of variance (ANOVA), effect of age: $F$=1.53, p=0.22; region by age interaction: $F$=1.44, p=0.49. Sample sizes: 11 adolescent, 9 adult (**E**) The average width of the area that dopamine axons occupy increased significantly from adolescence to adulthood, revealing that there is an increase in the total number of fibres passing to the medial prefrontal cortex during this period. Mixed-effects ANOVA, effect of age: $F$=9.45, p=0.0021; region by age interaction: $F$=5.74, p=0.057. Sample sizes: 11 adolescent, 9 adult (**F**) In order to quantify the Netrin-1-positive cells along the TH+ fibre pathway, the pathway was contoured in each region, and a contour of equal area was placed medial to the dopamine pathway as a negative control. Scale bar = 200 µm. (**G**) Using quantitative stereology, Netrin-1-positive cell density was determined along and adjacent to the pathway for each region. Red fluorescence indicates immunostaining for Netrin-1. (**H**) In adolescent mice there are more Netrin-1-positive cells along the fibres expressing TH ('TH+') than medial to them ('TH-'). This is what we refer to as the 'Netrin-1 pathway'. Along the pathway, there is a significant increase in Netrin-1-positive cell density in regions closer to the medial prefrontal cortex, the innervation target. Mixed-effects ANOVA, effect of TH: $F$=105, p<0.0001. Effect of region: $F$=9.74, p=0.021. A post hoc Tukey test revealed a difference (p=0.029) between the densities of the lateral septum and infralimbic cortex, but only within the dopamine pathway. Sample size: 8 (**I**) In adult mice the Netrin-1 pathway is maintained, however there is no longer an increasing density of Netrin-1-positive cells toward the medial prefrontal cortex. Mixed-effects ANOVA, effect of TH: $F$=54.56, p<0.0001. Effect of region: $F$=1.22, p=0.75. Sample size: 8 (**J**) The virus injection location within the mouse brain. A Netrin-1 knockdown virus, or a control virus, was injected into the dopamine pathway at the level of the dorsal peduncular cortex. (**K**) Our experimental timeline: at the onset of adolescence a Netrin-1 knockdown virus, or a control virus, was injected in wild-type mice. In adulthood the mice were sacrificed and stereological measurements taken. (**L**) TH+ varicosity density was quantified in the region below the injection site, the lateral septum, and in the region above the injection site, the infralimbic cortex. There was a significant decrease in TH+ varicosity density only in the infralimbic cortex. Mixed-effects ANOVA, virus by region interaction: $F$=16.41, p<0.0001. Sample sizes: knockdown 11, control 8 (**M**) The experimental set-up of the final (test) stage of the Go/No-Go experiment. A mouse that has previously learned to nose-poke for a reward in response to a visual cue (illuminated nose-poke hole) must now inhibit this behaviour when the visual cue is paired with an auditory cue (acoustic tone). (**N**) Mice injected with the Netrin-1 knockdown virus show improved action impulsivity compared to controls; they incur significantly fewer commission errors across the Go/No-Go task. Mixed-effects ANOVA, effect of day: $F$=68.32, p<0.0001. Day by virus interaction: $F$=9.00, p=0.0027. A sigmoidal curve is fit to each group of mice to determine how the two groups differ. Points indicate group means and error bars show standard error means. Sample sizes: knockdown 10, control 10 (**O**) During the first days of Go/No-Go testing, both groups incur commission errors with high frequency, but the Netrin-1 knockdown group has fewer errors than the control group (t-test, t=5.18, p<0.0001). (**P**) The ED50 – the inflection point in each sigmoidal curve – does not differ between groups, indicating that all mice improve their ability to inhibit their behaviour at around the same time (t-test, t=0.97, p=0.35). (**Q**) Mice microinfused with the Netrin-1 knockdown virus incur substantially fewer commission errors in the last days of the Go/No-Go task compared to mice injected with the control virus (t-test, t=12.38, p<0.0001). For all barplots, bars indicate group means and error bars show standard error means.

Next, we focussed on Netrin-1, a secreted protein that acts as a guidance cue to growing axons and is important for dopamine axon targeting in the nucleus accumbens in adolescence (*Cuesta et al., 2020*). Using unbiased stereology, we quantified the number of Netrin-1-expressing cell bodies along the dopamine axon route, and in an adjacent medial region as a control (*Figure 1F and G*). We found that in adolescence there are more Netrin-1-positive neurons within the dopamine axon route than adjacent to it. Furthermore, along the axon route the density of Netrin-1-positive cells increases toward the medial prefrontal cortex, forming a dorsoventral gradient (*Figure 1H*). In adulthood, there remains a higher density of Netrin-1-positive cells along the dopamine route compared to the adjacent region, however the dorsoventral gradient is no longer present (*Figure 1I*).

To determine if Netrin-1 along the dopamine axon route is necessary for axon navigation, we silenced Netrin-1 expression in the dorsal peduncular cortex, the transition region between the septum and the medial prefrontal cortex, at the onset of adolescence (*Figure 1J and K*). In adulthood, we quantified the number of dopamine axon terminals in the regions below and above the injection site. Silencing Netrin-1 did not alter dopamine terminal density below the injection, in the lateral septum (*Figure 1L*). In the infralimbic cortex, which is the first prefrontal cortical region the axons reach after the injection site, terminal density was reduced in the Netrin-1 knockdown group compared to controls (*Figure 1L*). The knockdown appears to erase the Netrin-1 path to the prefrontal cortex, resulting in dopamine axons failing to reach their correct innervation target.

It remains unknown exactly what types of cells are expressing Netrin-1 along the dopamine axon route, and how this expression is regulated to produce the Netrin-1 gradients that guide the dopamine axons. It also remains unclear where the misrouted axons end up in adulthood. Future experiments aimed at addressing these questions will provide further valuable insight into the nature of the

'Netrin-1 pathway'. Nonetheless, our results allow us to conclude that Netrin-1 expressing cells 'pave the way' for dopamine axons growing to the medial prefrontal cortex.

We next examined how the Netrin-1 pathway may be important for behaviour. Dopamine input to the prefrontal cortex is a key factor in the transition from juvenile to adult behaviours that occurs in adolescence. We hypothesised that cognitive processes involving mesocortical dopamine function would be altered when these axons are misrouted in adolescence. To test our hypothesis, we used the Go/No-Go behavioural paradigm. This test quantifies inhibitory control, which matures in parallel with the innervation of dopamine axons to the prefrontal cortex in adolescence (*Casey et al., 2008*; *Klune et al., 2021*; *Luna et al., 2015*; *Paus, 2005*; *Reynolds and Flores, 2021*; *Spear, 2000*), and it is impaired in adolescent-onset disorders like depression and schizophrenia (*Catts et al., 2013*; *Clementz et al., 2016*; *McTeague et al., 2016*; *Millan et al., 2012*).

At the onset of adolescence, we injected the Netrin-1 silencing, or a scrambled control virus, bilaterally into the dorsal peduncular cortex; in adulthood we tested the mice in the Go/No-Go task. This paradigm first involves discrimination learning and reaction time training (*Appendix 1 - Supplementary Analysis 1*), followed by a Go/No-Go test consisting of 'Go' trials where mice respond to a cue as previously trained and 'No-Go' trials where mice must abstain from responding to the cue (*Figure 1M*). Correct responses to both trial types are reinforced with a food reward. We quantified the percent of 'No-Go' trials where the mice incorrectly responded to the cue ('Commission Errors') and the percent of 'Go' trials where the mice correctly responded ('Rewards' or 'Hits'; *Appendix 1 - Supplementary Analysis 2*). The ability of mice to respond correctly overall to both trial types is quantified as the Correct Response Rate (*Appendix 1 - Supplementary Analysis 3*; *Reynolds et al., 2018*; *Vassilev et al., 2021*; *Reynolds et al., 2023*).

Mice injected with the Netrin-1 silencing virus differed from controls in their performance during 'No-Go' trials. As the mice learn to withhold their responses over the course of the test, the number of commission errors they made in No-Go trials decreased in a sigmoidal fashion (*Figure 1N*). The upper and lower asymptotes of the sigmoidal curve quantify the number of commission errors committed during early and late test days, respectively, while the inflection point (ED50) indicates when mice start improving their ability to inhibit their behaviour. At the start of the Go/No part of the task, the Netrin-1 silencing group make slightly fewer commission errors (*Figure 1O*) than control groups, although both groups begin to improve in the No-Go task at around the same time (*Figure 1P*). However, the Netrin-1 silencing group achieved a substantially higher level of behavioural inhibition, quantified as a lower percentage of commission errors in the last test days (*Figure 1Q*), indicating an improved ability to withhold their behaviour on cue. These behavioural results demonstrate that the maturation of action impulsivity is sensitive to the organisation of the ventro-dorsal Netrin-1 path that guides mesocortical dopamine axon growth. Deviations in this route associate with striking changes in the cognitive development that is characteristic of adolescence. In this case, the deviation leads to improved action impulsivity, suggesting that these dopamine axons may end up ectopically innervating a forebrain region other than the medial prefrontal cortex, enhancing cognitive control.

## Part 2: UNC5C expression coincides with the onset of adolescence

When axons leave the nucleus accumbens during adolescence, they follow a Netrin-1 'path' through intermediate brain regions to reach their intended innervation target. However, only a small subset of the dopamine axons that have reached the nucleus accumbens by early adolescence leave; the vast majority stay and form connections in the accumbens throughout life (*Reynolds et al., 2018*). The 'decision-making' process of whether to 'stay' (in the accumbens) or 'go' (to the cortex via the Netrin-1 path) happens during a narrow developmental window at the onset of adolescence (*Reynolds et al., 2019*). It remains unknown how the timing of this process is determined.

In adolescence, dopamine neurons begin to express the repulsive Netrin-1 receptor UNC5C, particularly when mesolimbic and mesocortical dopamine projections segregate in the nucleus accumbens (*Manitt et al., 2010*; *Reynolds et al., 2018*). In contrast, dopamine axons in the prefrontal cortex do not express UNC5c, except in very rare cases (*Appendix 1 - Supplementary Analysis 4*). In adult male mice with *Unc5c* haploinsufficiency, there appears to be ectopic growth of mesolimbic dopamine axons to the prefrontal cortex (*Auger et al., 2013*). This miswiring is associated with alterations in prefrontal cortex-dependent behaviours (*Auger et al., 2013*).

Using immunohistochemistry, we assessed the expression of UNC5C on nucleus accumbens dopamine axons across development. In male mice, we found little expression of UNC5C on dopamine axons at the onset of adolescence (*Figure 2A*), while we did find UNC5C expression on dopamine axons in adults (*Figure 2B*). Remarkably, when we assessed this in females, we found dopamine axons already expressing UNC5C in the nucleus accumbens at the onset of adolescence (*Figure 2D*), similar to adult females (*Figure 2E*), indicating that the onset of UNC5C expression on dopamine axons in the nucleus accumbens is sexually dimorphic, with an earlier emergence in females. We examined the nucleus accumbens in pre-adolescent female mice and indeed found little UNC5C expression on dopamine axons (*Figure 2C*). The onset of UNC5C expression in mesocorticolimbic dopamine axons is therefore peri-adolescent but occurs earlier in females than in males, consistent with the earlier emergence of adolescence in female rodents and the earlier onset of adolescence and puberty in humans (*Wolf and Long, 2016*). Differences in the precise timing of dopamine innervation to the PFC in adolescence have been suggested by findings reported in male and female rats (*Willing et al., 2017*).

## Part 3: Environmental control of the timing of adolescence

We hypothesise that at the emergence of adolescence, UNC5C expression by dopamine axons in the nucleus accumbens signals the initiation of the growth of dopamine axons to the prefrontal cortex. We therefore examined whether the developmental timings of UNC5C expression and dopamine innervation of the prefrontal cortex are similarly affected by an environmental cue known to delay pubertal development in seasonal species.

Siberian hamsters (*Phodopus sungorus*) regulate many aspects of their behaviour and physiology to meet the changing environmental demands of seasonality (*Paul et al., 2008*; *Stevenson et al., 2017*). In winter, they increase the thickness of their fur, exchange their brown summer coats for white winter ones, and undergo a daily torpor to conserve energy (*Scherbarth and Steinlechner, 2010*). In addition, adults suppress reproduction and juveniles delay puberty (*Pévet, 1988*; *Yellon and Goldman, 1984*), including developmental changes in gonadotropin releasing hormone neurons in the hypothalamus (*Buchanan and Yellon, 1991*; *Heywood and Yellon, 1997*). Reproductive organ development is delayed as part of pubertal postponement (*Darrow et al., 1980*; *Ebling, 1994*; *Timonin et al., 2006*). This seasonal plasticity is regulated by long or short periods of daylight (*Heldmaier and Steinlechner, 1981*; *Hoffmann, 1978*) and raises the possibility that aspects of adolescent development are sensitive to these environmental cues (*Paul et al., 2018*; *Walker et al., 2017*). To our knowledge, adaptive variation in the timing of adolescent neural development has never been recorded in any animal.

Here, we tested whether daylength regulates *when* dopamine axons grow to the cortex, and whether the timing of UNC5C expression in the nucleus accumbens and adolescent changes in behaviour are similarly affected.

### The seasonality of adolescence

Male hamsters were examined at three ages: 15 days of age (±1), 80 days of age (±10), and 215 days of age (±20). We compared the density of the dopamine innervation to the medial prefrontal cortex in male hamsters housed under lighting conditions that replicate summer daylengths (long days, short nights) or winter daylengths (short days, long nights) (*Figure 3A and B*). We will refer to these two groups as 'summer hamsters' and 'winter hamsters' to emphasise the natural stimulus we are replicating in the laboratory environment. We confirmed that puberty is delayed in male winter hamsters compared to summer hamsters in the present experiment by measuring their gonadal weights across ages (*Appendix 1 - Supplementary Analysis 5*).

In male summer hamsters, dopamine input density to the prefrontal cortex increases during adolescence, after 15 days of age and before 80 days of age (*Figure 3C*), consistent with dopamine axon growth in mice (*Manitt et al., 2013*; *Manitt et al., 2011*; *Reynolds et al., 2018*). Prefrontal cortex dopamine innervation in summer hamsters continues to increase after 80 days of age (*Figure 3C*).

In male winter hamsters, dopamine innervation to the prefrontal cortex is delayed until after 80 days, which coincides with their delayed pubertal onset (*Figure 3D*, *Appendix 1 - Supplementary Analysis 5*). This demonstrates that an environmental cue can determine the timing of adolescent brain development.

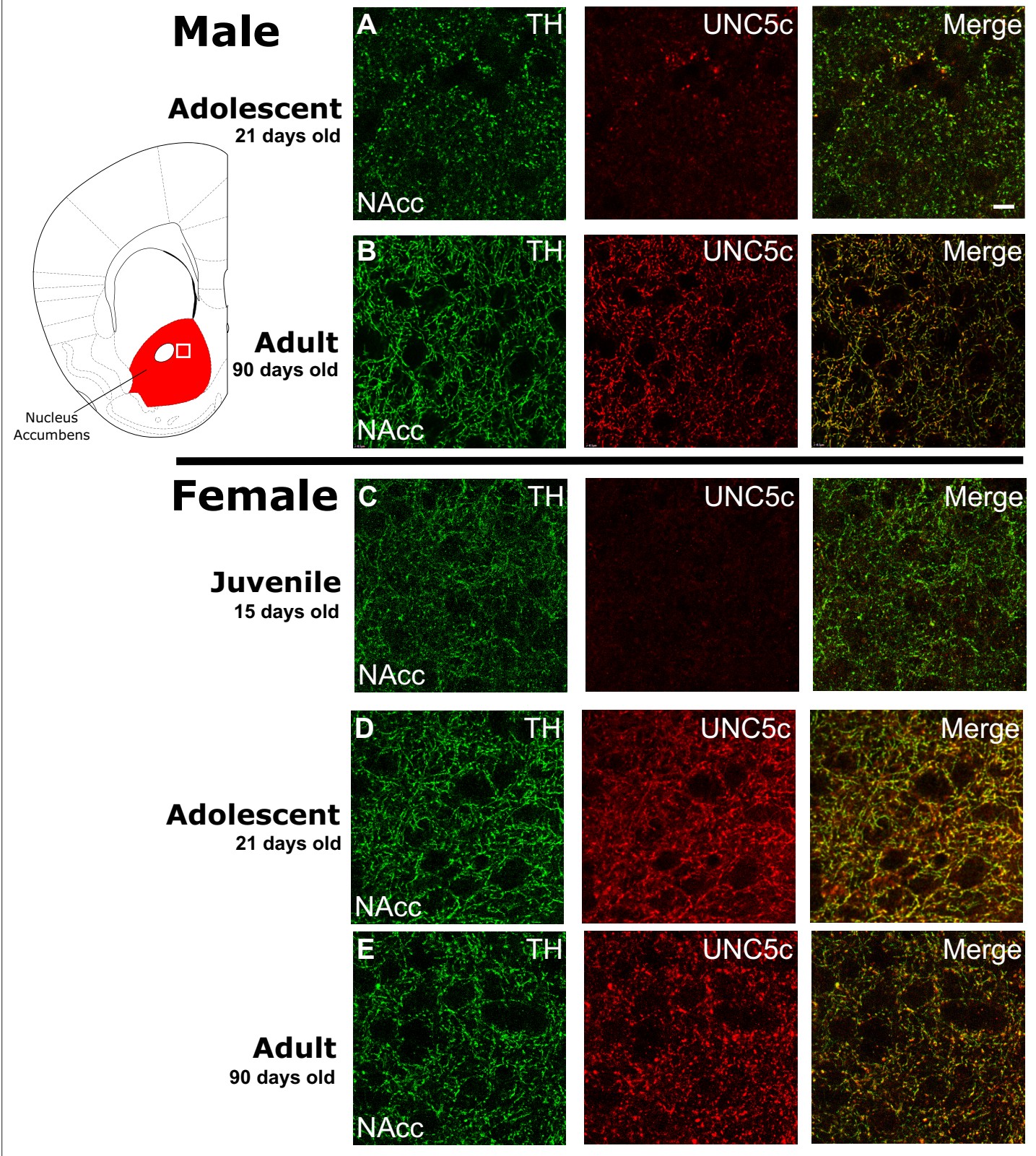

**Figure 2.** The age of onset of UNC5C expression by dopamine axons in the nucleus accumbens of mice is sexually dimorphic. Images are representative of observed immunofluorescence patterns in the nucleus accumbens (approx. location highlighted as a white square in the coronal mouse brain section plate 19, modified from *Paxinos and Franklin, 2013*). No qualitative differences were noted between the shell and core of the nucleus accumbens. For each row, six individuals were sampled. In males (**A–B**), UNC5C expression on dopamine fibres (here identified by

*Figure 2 continued on next page*

*Figure 2 continued*

immunofluorescent staining for tyrosine hydroxylase [TH]) in the nucleus accumbens appears during adolescence. (**A**) At the onset of adolescence (21 days of age) dopamine fibres do not express UNC5C. Scale bar = 10 μm. (**B**) By adulthood (90 days of age), dopamine fibres express UNC5C. In females (**C–E**), UNC5C expression on dopamine fibres in the nucleus accumbens appears prior to adolescence. (**C**) In juvenile (15 days of age) mice, there is no UNC5C expression on dopamine fibres. (**D**) By adolescence, dopamine fibres express UNC5C. (**E**) In adulthood, dopamine fibres continue to express UNC5C.

We then examined UNC5C expression by dopamine axons in the nucleus accumbens in male summer and winter hamsters across age classes. UNC5C expression was apparent only after the onset of adolescence in summer hamsters (*Figure 3E, F, and G*), as observed in male mice. However, UNC5C expression was delayed in male winter hamsters – this group did not show UNC5C expression in dopamine axons in the nucleus accumbens until after 80 days of age (*Figure 3H, I, and J*). This aligns with the delayed timing of mesocortical dopamine axon growth and pubertal onset in male winter hamsters and demonstrates that the emergence of UNC5C is a marker of adolescent onset in male mice.

A behavioural characteristic of adolescence is increased willingness to enter a novel environment, a behaviour that assumes an increased amount of risk (*Arrant et al., 2013*; *Lynn and Brown, 2009*). To measure this, we used the light/dark test (*Bourin and Hascoët, 2003*). Time spent in the light compartment is dopamine-dependent (*Bahi and Dreyer, 2019*; *Gao and Cutler, 1993*) and peaks in adolescence (*Arrant et al., 2013*). We will refer to this behaviour as 'risk taking'. We assessed the developmental profile of risk taking in the light/dark box test in summer and winter hamsters across adolescence. In male summer hamsters, the risk taking increases across adolescence, peaks around 50 days, then subsequently declines (*Figure 3K*). However, the adolescent increase in risk taking is protracted in winter hamsters: across the age range examined we observe a gradual, consistent increase in risk taking rather than a peak and decline.

We next assessed a cohort of 215-day-old hamsters, for which both summer and winter male hamsters have undergone puberty and exhibit high levels of dopamine innervation of the prefrontal cortex (*Figure 3C, D, G, and J*, *Appendix 1 - Supplementary Analysis 5*). In these hamsters, we find no difference in risk taking between the male summer and winter groups (*Figure 3L*), demonstrating that, after 80 days, risk taking begins to decline in male winter hamsters and that by 215 days it has declined to the same level as in summer hamsters. Male hamsters raised under summer-mimicking long days and winter-mimicking short days both ultimately make the transition to the adult behavioural phenotype.

We also examined a second behaviour, novel object investigation. We note similar, but not identical, developmental patterns in behaviour. Both male summer and winter hamsters show peaks in novel object exploration around 50 days old, however the developmental shifts in behaviour around that peak are significantly more substantial in the summer males compared to the winter males (Appendix 1 - Supplementary Analysis 8). In 215-day-old hamsters, there is no difference in novel object exploration between summer and winter males (Appendix 1 - Supplementary Analysis 9).

## An extraordinary case of decoupling puberty and adolescence

In parallel with males, we conducted equivalent experiments in female hamsters (*Figure 4A and B*). Under a summer-mimicking daylength, dopamine innervation to the medial prefrontal cortex increases between 15 and 80 days of age, similar to male summer hamsters (*Figure 4C*). There is no further increase in innervation density after 80 days of age, consistent with earlier adolescent development in females observed in other rodent species (*Juraska and Willing, 2017*; *Kopec et al., 2018*; *Reynolds and Flores, 2021*; *Spear, 2000*; *Westbrook et al., 2018*). We confirmed that puberty is delayed in female winter hamsters compared to summer hamsters by measuring their uterine weights (*Appendix 1 - Supplementary Analysis 6*) and vaginal opening (*Appendix 1 - Supplementary Analysis 7*) across ages.

When housed under a winter-mimicking daylength, dopamine input density in the prefrontal cortex of female hamsters is *not* delayed as in males, but rather reaches adult levels prior to 15 days of age (*Figure 4D*). We replicated this unexpected finding in a separate, independent cohort of female winter hamsters (*Figure 4E*). This surprising result shows an intervention that accelerates adolescent cortical development.

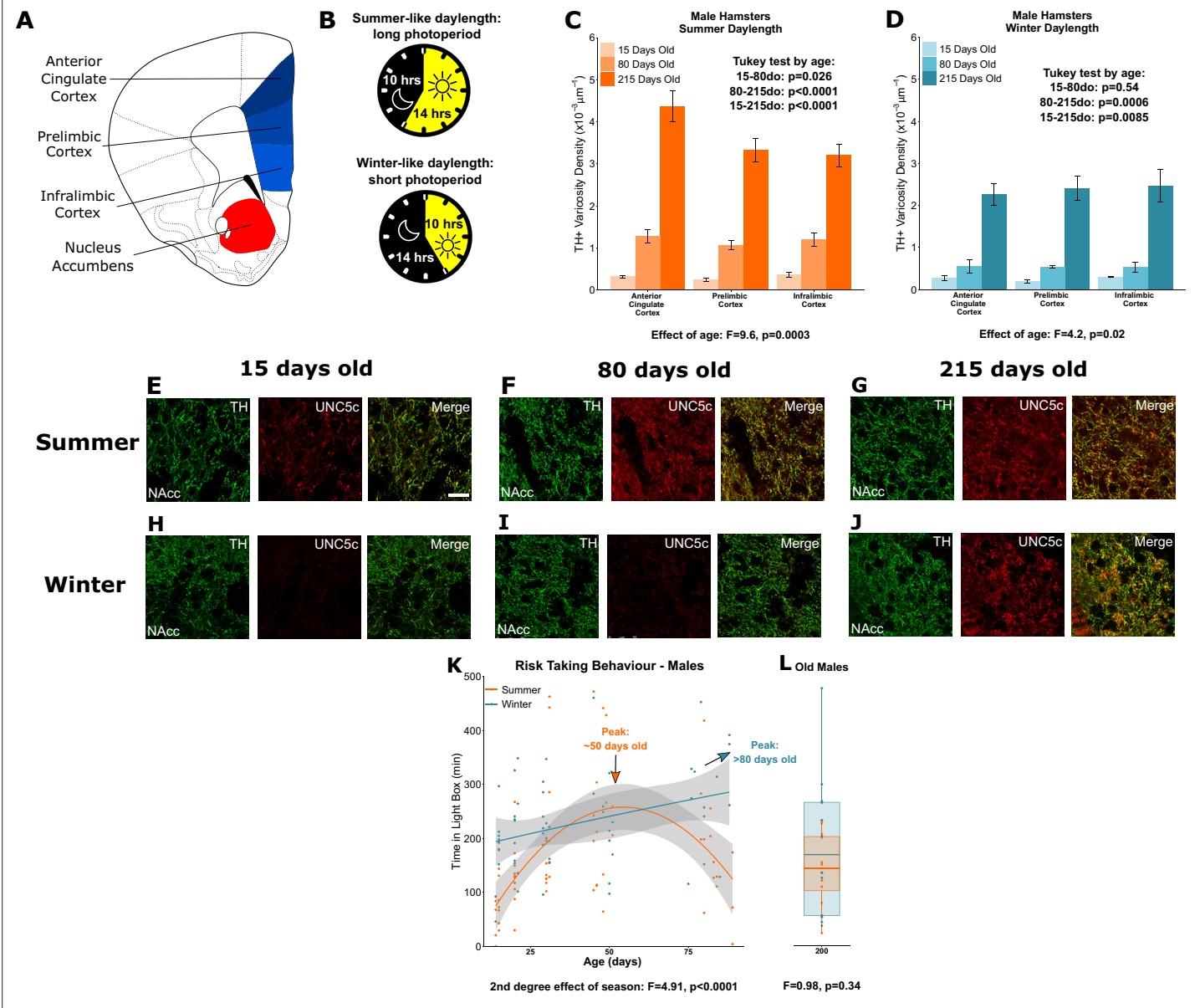

**Figure 3.** Plasticity of adolescent development in male Siberian hamsters according to seasonal phenotype. All results illustrated in this figure refer to results in male hamsters. (**A**) Dopamine innervation was quantified in three subregions of the medial prefrontal cortex, highlighted in blue. UNC5C expression was examined in the nucleus accumbens, highlighted in red. Line drawing of a coronal section of the mouse brain was derived from plate 19 of *Paxinos and Franklin, 2013*. (**B**) Hamsters were housed under either summer-mimicking long days and short nights ('summer hamsters') or winter-mimicking short days and long nights ('winter hamsters'). (**C**) In male hamsters housed under a summer-mimicking daylength there is an increase in dopamine varicosity density in the medial prefrontal cortex between 15 and 80 days of age. Mixed-effects analysis of variance (ANOVA), effect of age: $F$=9.6, p=0.000255. Tukey test, 15–80 days old (do): p=0.026; 80–215do: p<0.0001; 15–215do: p<0.0001. Sample sizes: 15-days-old 8, 80-days-old 8, 215-days-old 10 (**D**) In male hamsters housed under a winter-mimicking daylength there is no increase in dopamine varicosity density until hamsters have reached 215 days of age. Mixed-effects ANOVA, effect of age: $F$=4.17, p=0.0205. Tukey test, 15–80do: p=0.54; 80–215do: p=0.0006; 15–215do: p=0.0085. Sample sizes: 15-days-old 4, 80-days-old 8, 215-days-old 8 (**E**) At 15 days of age, dopamine axons (here identified by immunofluorescent staining for tyrosine hydroxylase [TH]) in the nucleus accumbens of male summer daylength hamsters largely do not express UNC5C. Scale bar = 20 μm (bottom right). (**F–G**) At 80 (**F**) and 215 (**G**) days of age, dopamine axons in the nucleus accumbens express UNC5C. (**H–I**) At 15 (**H**) and 80 (**I**) days of age, dopamine axons in the nucleus accumbens of male winter hamsters largely do not express UNC5C. (**J**) By 215 days of age there is UNC5C expression in dopamine axons in the nucleus accumbens of male winter hamsters. (**E–J**) Representative images of the nucleus accumbens shell, six individuals were examined per group. (**K**) Male hamsters house under a summer-mimicking daylength show an adolescent peak in risk taking in the light/dark box apparatus. Those raised under a winter-mimicking photoperiod show a steady increase in risk taking over the same age range. Arrows indicate the ages at which risk-taking peaks in summer (orange) and winter (blue) hamsters. Polynomial regression, effect of season: $F$=3.551, p=0.00056. Curves show

*Figure 3 continued on next page*

*Figure 3 continued*

polynomial functions, shaded areas show uncertainty in the functions. Sample sizes: summer 66, winter 57 (**L**) In male hamsters, at 215 days of age, there is no difference in risk taking between hamsters raised under summer and winter photoperiods. t-Test, effect of season: t=0.975, p=0.341. The central line through each box indicates the group mean, the upper and lower bounds of each box indicate the third and first quartiles respectively, and the whiskers indicate the maximum and minimum values. Sample sizes: 12 summer, 12 winter. For all barplots, bars indicate group means and error bars show standard error means.

We then measured dopamine axon density in female winter hamsters at two earlier ages: 10 and 15 days of age. Dopamine innervation increases during this period (*Figure 4F*), well before normal adolescence and long before pubertal development. This is an extraordinary phenomenon: a key marker of adolescent neurodevelopment is accelerated and dissociated from puberty in female hamsters raised under winter-mimicking short days (*Appendix 1 - Supplementary Analyses 6,7*).

The early increase in prefrontal cortex dopamine terminals in winter females is followed by a dramatic reduction between 80 and 215 days of age (*Figure 4D and E*). This overlaps with the delayed timing of puberty in these females (*Butler et al., 2007*; *Appendix 1 - Supplementary Analyses 6,7*). Synaptic pruning in the cortex is a well-known component of adolescent neural development across species (*Huttenlocher, 1984*; *Koss et al., 2014*; *Petanjek et al., 2011*). Under normal conditions, the effect of pruning on dopamine synapses is likely masked by the growth of new dopamine axons to the prefrontal cortex (*Manitt et al., 2013*; *Manitt et al., 2011*; *Reynolds et al., 2018*). In the case of female winter hamsters, we hypothesise that the growth of dopamine axons to the prefrontal cortex occurs early while synaptic pruning, including dopamine synapses, appears to occur later. This leads to a remarkable dissociation between two cortical developmental processes that are normally simultaneous, the behavioural implications of which are unclear.

If the developmental onset of UNC5C expression determines the timing of dopamine innervation of the prefrontal cortex, then onset of UNC5C expression should also be advanced in female winter hamsters. Hence, we examined UNC5C expression at the same ages as we examined dopamine axon growth in female hamsters. At 10 and 15 days of age, UNC5C expression is present *only* in the winter hamsters (*Figure 4G, H, K, and L*), but at 80 and 215 days of age, UNC5C expression is apparent in both summer and winter hamsters (*Figure 4I, J, M, and N*).

We used the light/dark box test to examine potential risk-taking implications of the extraordinary developmental trajectory we observed in the prefrontal cortex of female hamsters. In female summer and winter hamsters, the adolescent increase and peak in risk taking occurs between the ages of 15 and 80 days, as it does in summer daylength males (*Figure 4O*). However, contrary to what we would expect, the peak in winter females is delayed compared to summer females. A delayed peak is also observed for winter females in novel object investigation (Appendix 1 - Supplementary Analysis 10). When we assessed an independent cohort of 215-day-old female hamsters, we found no difference in risk taking (*Figure 4P*) or novel object investigation (Appendix 1 - Supplementary Analysis 11) between groups, indicating that, like males, female summer and winter hamsters both eventually reach the same adult level of risk taking.

In both sexes, hamsters housed under a summer-mimicking daylength showed an adolescent peak in risk taking at an age that we would predict based on results from other rodents (*Arrant et al., 2013*; *Pietropaolo et al., 2004*; *Tanaka, 2015*). When raised under a winter-mimicking daylength, hamsters of either sex show a protracted peak in risk taking. In males, it is delayed beyond 80 days of age, but the delay is substantially less in females. This is a counterintuitive finding considering that dopamine development in winter females appears to be accelerated. Our interpretation of this finding is that the timing of the risk-taking peak in females may reflect a balance between different adolescent developmental processes. The fact that dopamine axon growth is accelerated does not imply that all adolescent maturational processes are accelerated. Some may be delayed, e.g., those that induce axon pruning in the cortex. The timing of the risk-taking peak in winter female hamsters may therefore reflect the amalgamation of developmental processes that are advanced with those that are delayed – producing a behavioural effect that is timed somewhere in the middle. Disentangling the effects of different developmental processes on behaviour will require further experiments

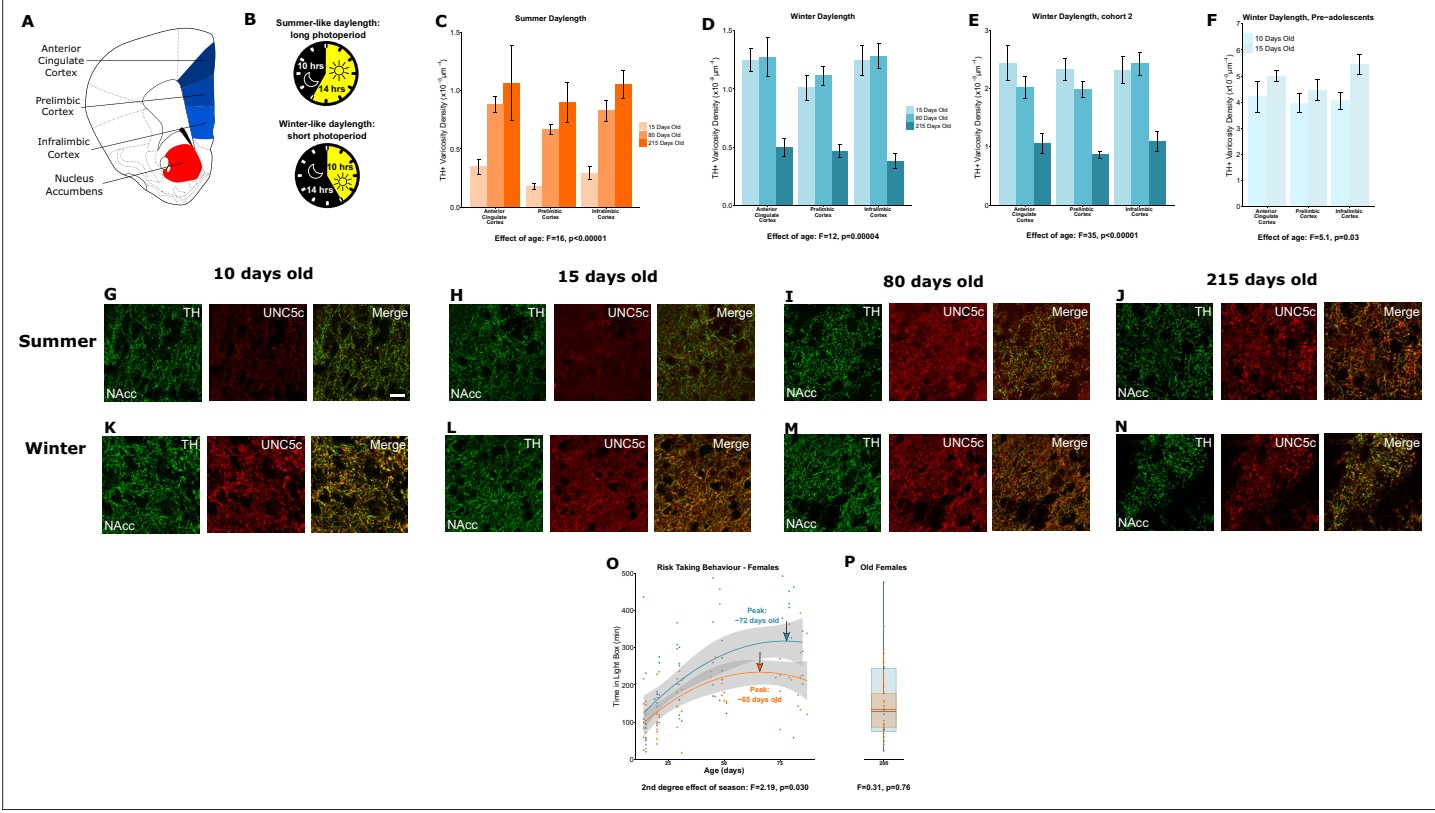

**Figure 4.** Plasticity of adolescent development in female Siberian hamsters according to seasonal phenotype. All results illustrated in this figure refer to results in female hamsters. (**A**) Dopamine innervation was quantified in three subregions of the medial prefrontal cortex, highlighted here in blue. UNC5C expression was examined in the nucleus accumbens, highlighted in red. Line drawing of a coronal section of the mouse brain was derived from *Paxinos and Franklin, 2013*. (**B**) Hamsters were housed under either a summer-mimicking or winter-mimicking daylength. (**C**) In female hamsters housed under a summer daylength dopamine varicosity density in the medial prefrontal cortex increases between 15 and 80 days of age. Mixed-effects analysis of variance (ANOVA), effect of age: $F=16.72$, $p<0.0001$. Sample sizes: 15-days-old 6, 80-days-old 8, 215-days-old 4 (**D**) In female hamsters housed under a winter daylength there is no increase in dopamine varicosity density post-adolescence. Instead, there is a steep decline in density between 80 and 215 days of age. Mixed-effects ANOVA, effect of age: $F=12.33$, $p=0.000043$. Sample sizes: 15-days-old 8, 80-days-old 8, 215-days-old 8 (**E**) As our results in panel D were unexpected, we replicated them with a second cohort of hamsters and found qualitatively identical results. Mixed-effects ANOVA, effect of age: $F=34.871$, $p<0.0001$. 15-days-old 8, 80-days-old 8, 215-days-old 7 (**F**) To try and determine when dopamine varicosities innervate the medial prefrontal cortex, we examined a cohort of 10- and 15-day-old hamsters. We found that varicosity density increases in the medial prefrontal cortex during this time, indicating that dopamine innervation to the medial prefrontal cortex is accelerated in female winter hamsters. Mixed-effects ANOVA, effect of age: $F=5.05$, $p=0.03$. Sample sizes: 10-days-old 10, 15-days-old 8 (**G–H**) In 10- and 15-day-old female summer hamsters there is little UNC5C expression in nucleus accumbens dopamine axons (here identified by immunofluorescent staining for tyrosine hydroxylase [TH]). Sample size: 4 (panel G) or 6 (panel H). (**I–J**) By 80 days of age (panel I), and continuing at 215 days of age (panel J), dopamine axons in the nucleus accumbens express UNC5C in female summer hamsters. Sample sizes: 6. Scale bar = 20 μm (panel G bottom right). (**K–N**) At all ages which winter female hamsters were examined, dopamine axons in the nucleus accumbens express UNC5C in winter female hamsters. Sample sizes: 4 (panel K) or 6 (panels L–N). (**O**) In female hamsters, those raised under summer and winter daylengths both show an increase in risk taking over time. The winter hamsters peak later compared to the summer daylength hamsters. Arrows indicate the ages at which risk taking peaks in summer (orange) and winter (blue) hamsters. Polynomial regression, effect of season: $F=3.305$, $p=0.00126$. Curves show polynomial functions, shaded areas show uncertainty in the functions. Sample sizes: summer 66, winter 61 (**P**) In female hamsters, at 215 days of age, there is no difference in risk taking between hamsters raised under summer and winter photoperiods. t-Test, effect of season: $t=0.309$, $p=0.76$. The central line through each box indicates the group mean, the upper and lower bounds of each box indicate the third and first quartiles respectively, and the whiskers indicate the maximum and minimum values. Sample sizes: 15 summer, 12 winter. For all barplots, bars indicate group means and error bars show standard error means.

in hamsters, including the direct manipulation of dopamine activity in the nucleus accumbens and prefrontal cortex.

## Conclusion

Here, we describe how the gradual growth of mesocortical dopamine axons marks adolescent development, and how this process uses guidance cues and is sensitive to sex and environment. Netrin-1

signalling provides the 'stay-or-go' 'decision making' conducted by dopamine axons that innervate the nucleus accumbens at the onset of adolescence (*Cuesta et al., 2020*; *Reynolds et al., 2023*). UNC5C expression by these dopamine axons marks the timing at which this decision is made. In mice, UNC5C expression coincides with sex differences in both adolescent and pubertal development. Females, which develop earlier, show earlier UNC5C expression in dopamine axons compared to males.

In hamsters, behavioural and developmental shifts in response to environmental cues occur in parallel with alterations in the timing of dopamine axon growth. As we show here, male hamsters raised under a winter-mimicking daylength delay not only puberty, but also adolescent dopamine and behavioural maturation. In contrast, female hamsters under identical conditions delay puberty but accelerate dopamine axon growth, a key marker of adolescent brain development. Behavioural shifts during adolescence appear to be delayed in these females, but less substantially than in male hamsters. Notably, under all conditions, the developmental timing of UNC5C expression corresponds to the timing of dopamine innervation of the prefrontal cortex.

In both mice and hamsters, the emergence of UNC5C expression coincides with the onset of dopamine axon growth to the prefrontal cortex, a key characteristic of the adolescent transition period. While previously we have shown that the Netrin-1 signalling in the nucleus accumbens is responsible for coordinating *whether* dopamine axons grow in adolescence (*Reynolds and Flores, 2021*; *Cuesta et al., 2020*; *Reynolds et al., 2023*), here we propose that Netrin-1 signalling is also key to determining *how* and *when* this marker of adolescence occurs.

## Methods

### Animals

All mouse (*Mus musculus*) experiments and procedures were performed in accordance with the guidelines of the Canadian Council of Animal Care and the McGill University/Douglas Hospital Research Centre Animal Care Committee. All mice were received from Charles River Canada and housed with same-sex littermates on a 12 hr light/dark cycle with ad libitum access to food and water. We used male mice for these experiments.

All Siberian hamster (*P. sungorus*) experiments and procedures were approved by the University at Buffalo, SUNY Institutional Animal Care and Use Committee. Hamsters were obtained from our colony (MJP), which was originally derived from animals generously provided by Dr. Brian Prendergast, University of Chicago, in 2015. Hamsters were housed with same-sex littermates in well-ventilated, light-proof environmental housing units that provided either a summer-mimicking long-day photoperiod (14:10 hr light:dark cycle) or a winter-mimicking short-day photoperiod (10:14 hr light:dark cycle); dim red light was present during the dark phase. Food and water were available ad libitum. Both male and female mice were used for these experiments.

### Tissue processing

Rodents were euthanised with an intraperitoneal injection of 50 mg/kg ketamine, 5 mg/kg xylazine, and 1 mg/kg acepromazine. They were then perfused intracardially with 10 IU/mL heparinised saline (mice) or physiological saline (hamsters) followed by 4% paraformaldehyde. Both perfused solutions were pH-adjusted to between 7.2 and 7.4 with dilute hydrochloric acid and sodium hydroxide. After perfusion, brains were dissected from the skull, placed in fixative solution overnight at 4°C and then stored in phosphate-buffered saline at 4°C. Brains were cut coronally into 30 μm (hamster) or 35 μm (mouse) thick sections on a vibratome.

### Immunohistochemistry

Every second section (mouse) or third section (hamster) was processed for immunofluorescence as we have described previously (*Salameh et al., 2018*).

For experiments in mouse tissue requiring only tyrosine hydroxylase (TH) staining, we used a rabbit anti-TH (1:1000 dilution, product #AB152; Millipore) antibody as the primary antibody and an Alexa Fluor (AF) 594-conjugated donkey anti-rabbit antibody (1:500 dilution, product #711585152, Jackson Laboratories) as the secondary antibody. We and others have shown that the TH antibody used in these studies labels dopamine axons but rarely labels norepinephrine axons within the regions of interest (*Manitt et al., 2013*; *Manitt et al., 2011*; *Miner et al., 2003*; *Reynolds et al., 2022*).

To examine hamster sections for TH only, a 3,3'-diaminobenzidine (DAB) staining protocol was used. First, we performed antigen retrieval using heated (70°C) citrate buffer (0.05 M) followed by glycine (0.1 M). We used a mouse anti-TH antibody (1:22,000 dilution, product #T1299, Sigma), followed by secondary staining using a DAB staining kit (product #SK4100, Vector Laboratories).

To detect both TH and Netrin-1, we used a mouse anti-TH antibody (1:1000 dilution, product #MAB318; Millipore) and a rabbit Netrin-1 antibody (1:500 dilution, product #ab126729, abcam) as primary antibodies. We use citrate buffer and sodium dodecyl substrate antigen recovery methods to strengthen the Netrin-1 signal as previously described (*Salameh et al., 2018*). We used AF488-conjugated donkey anti-mouse antibody (1:500 dilution, product #715545150, Jackson Laboratories) and the AF594-conjugated donkey anti-rabbit antibody (as above) as secondary antibodies.

To detect both TH and UNC5C, we first used the rabbit anti-TH antibody (1:1000 dilution) and a mouse anti-UNC5C antibody (1:100 dilution, provided by Dr. Guofa Liu) as primary antibodies and an AF488-conjugated donkey anti-rabbit (1:500 dilution, product #711545152, Jackson Laboratories) and AF594-conjugated donkey anti-mouse (1:500 dilution, product #711585152, Jackson Laboratories) secondary antibodies. For these experiments we used the same antigen retrieval methods as with our immunohistochemistry staining for Netrin-1 described above and in *Salameh et al., 2018*.

To replicate our results with a commercially available antibody, we used a goat anti-UNC5C antibody (1:200 dilution, product #NBP1-37002, NOVUS Biologicals) along with the rabbit anti-TH antibody (1:500 dilution) as primary antibodies. We used the AF488-conjugated donkey anti-rabbit (as above) and AF594-conjugated donkey anti-goat (1:500 dilution, product #705585003, Jackson Laboratories) antibodies as secondary antibodies. For this experiment we used two variations on the standard immunohistochemistry protocol described in *Salameh et al., 2018*: Tris-buffered saline was used in place of phosphate-buffered saline, and commercial protein block and antibody diluent (both from Agilent) were used in place of a bovine serum albumin blocking solution.

In all immunochemistry experiments we stain for TH as a marker for dopamine in order to identify dopamine axons. Therefore, we pay great attention to the morphology and localisation of the fibres to avoid including in our study any fibres stained with TH antibodies that are not dopamine fibres. The fibres that we examine and that are labelled by the TH antibody show features indistinguishable from the classic features of cortical dopamine axons in rodents (*Berger et al., 1983*; *Berger et al., 1974*; *Van Eden et al., 1987*; *Manitt et al., 2011*), namely they are thin fibres with irregularly spaced varicosities, are densely packed in the nucleus accumbens, sparsely present only in the deep layers of the prefrontal cortex, and are not regularly oriented in relation to the pial surface. This is in contrast to rodent norepinephrine fibres, which are smooth or beaded in appearance, relatively thick with regularly spaced varicosities, increase in density toward the shallow cortical layers, and are in large part oriented either parallel or perpendicular to the pial surface (*Berger et al., 1983*; *Berger et al., 1974*; *Levitt and Moore, 1979*; *Miner et al., 2003*). Furthermore, previous studies in rodents have noted that only norepinephrine cell bodies are detectable using immunofluorescence for TH, not norepinephrine processes (*Miner et al., 2003*; *Pickel et al., 1975*; *Verney et al., 1982*), and we did not observe any norepinephrine-like fibres. Finally, a DAT-Cre approach was used to demonstrate that all axons that immunostain for TH in the forebrain are dopamine axons (*Caldwell et al., 2023*). We are not aware of any other processes in the forebrain that are known to be immunopositive for TH under any environmental conditions.

After immunofluorescence staining, sections were mounted on gel-coated slides and cover-slipped with a fluorescence-preserving mounting medium ('Vectashield' branded media, Vector Laboratories). Sections were either stained with DAPI prior to mounting or mounted with a DAPI-containing medium.

## Stereological analyses

For all experiments, contours were delineated on sections corresponding to plates 13–22 of the mouse brain atlas (*Paxinos and Franklin, 2013*) or plates 10–14 of the hamster brain atlas (*Morin and Wood, 2001*). The brain regions along the dopamine axon route from the nucleus accumbens to the prefrontal cortex consist of the lateral septum, dorsal peduncular cortex, and infralimbic cortex; the latter being the first medial prefrontal cortex subregion encountered along this route. The subregions of the medial prefrontal cortex in which dopamine varicosities were quantified in this study are the infralimbic cortex, prelimbic cortex, and anterior cingulate cortex. All regions were examined only

anterior to the genu of the corpus callosum. Counting was conducted bilaterally in mice and in the left hemisphere in hamsters.

Dopamine axon density along the route from the nucleus accumbens to the medial prefrontal cortex was determined using a modified stereological approach based on that described in *Kim et al., 2011*. The dense bundle of dopamine fibres that occurs along the lateral boundary of each region was traced at ×5 magnification with a Leica DM400B microscope and StereoInvestigator (Microbrightfield) software. Using the counting probe function of StereoInvestigator, a grid of 175 µm$^2$ was superimposed on each contour, starting at a random starting point within the contour. Unbiased counting frames (length = 25 µm, width = 10 µm) were placed in the top left corner of each grid square. Axons were counted if they crossed both the upper and lower boundaries of the counting frame. Counting was conducted at ×40 magnification using a counting depth of 10 µm and a guard zone of 5 µm. Counts were performed blind by a single individual (TO). Axon density was determined by dividing the total axon count by the width of the contour.

Netrin-1-positive cell bodies were used as the counting unit to examine Netrin-1 density along the dopamine axon route from the nucleus accumbens to the medial prefrontal cortex. The dense bundle of dopamine fibres that occurs along the lateral boundary of each region was traced at ×5 magnification with a Leica DM400B microscope and StereoInvestigator software. We also delineated a region of equal area directly medial to the fibre bundle, and we considered this the TH-negative subregion (*Figure 1F* of the main text). To determine the number of Netrin-1-positive cell bodies, we used the optical fractionator probe function of StereoInvestigator with a grid of 175 µm$^2$, an unbiased counting frame of 100 µm$^2$, a counting depth of 10 µm, and a guard zone of 2 µm. Counting was conducted at ×40 magnification using the standard counting protocol for quantifying discrete objects ('particle stereology'; *Howard and Reed, 2005*). Counts were performed blind by a single individual (SS). To determine the volume of each subregion we used the Cavalieri method in StereoInvestigator (*Howard and Reed, 2005*). The coefficient of error was below 0.1 for all measures. Cell density was determined by dividing the total count of cells by the volume of the subregion.

TH-positive varicosities were used as the counting unit to obtain a measure of dopamine presynaptic density because nearly every dopamine varicosity in the prefrontal cortex forms a synapse (*Séguéla et al., 1988*). Varicosities also represent sites where neurotransmitter synthesis, packaging, release, and reuptake most often occur (*Benes et al., 1996*). Stereology was conducted as previously described (*Manitt et al., 2011*; *Reynolds et al., 2022*). Contours of the dense TH-positive innervation in the medial prefrontal cortex were traced at ×5 magnification using a Leica DM400B microscope and StereoInvestigator software. To determine the number of TH-positive varicosities, we used the optical fractionator probe function of StereoInvestigator with a grid of 175 µm$^2$, a counting frame of 25 µm$^2$, a counting depth of 10 µm, and a guard zone of 5 µm. Counting was conducted at ×100 magnification using the standard counting protocol for quantifying discrete objects ('particle stereology'; *Howard and Reed, 2005*). Counts were performed blind by one individual per experiment (DH, AH, TO, or AD depending on the experiment). To determine the volume of each subregion we used the Cavalieri method in StereoInvestigator (*Howard and Reed, 2005*). The coefficient of error was below 0.1 for all measures. Varicosity density was determined by dividing the total count of varicosities by the volume of the subregion.

## Stereotaxic surgery

To experimentally knock down Netrin-1 along the dopamine axon route from the nucleus accumbens to the medial prefrontal cortex, we injected a Netrin-1 shRNA-expressing lentivirus or a scrambled control virus into the dorsal peduncular cortex.

Pre-designed and validated siRNA sequences (Ambion) were used to create shRNA (sequence GGAGCUCUAUAAGCUAUCA) by the addition of a standard hairpin loop (TTCAAGAGA) between the sense and antisense sequences. A scrambled control was created by rearranging the sequence order so that there was less than a 64% interaction rate. Active or control shRNA sequences were cloned into a pLentiLox 3.7 vector (Addgene, Plasmid #11795). Lentiviruses expressing the shRNAs and scrambled controls were prepared by the SPARC Biocentre lentiviral core facility (SickKids Hospital, Toronto, ON, Canada). For more details and validation information, see *Cuesta et al., 2020*.

21-day-old mice were anaesthetised with isoflurane (5% for induction and 2% for maintenance) and placed in a stereotaxic apparatus. Using Hamilton syringes, the shRNA-expressing lentivirus, or the

lentivirus expressing the scrambled control sequence, were microinfused bilaterally into the dorsal peduncular cortices stereotaxically using the coordinates: +2.00 mm anterior/posterior, –0.05 mm medial/lateral, and –3.45 mm dorsal/ventral relative to Bregma. A total of 0.5 µL of purified virus was delivered on each side at an injection rate of 0.08 µL/min, which was then followed by a 3 min pause to allow of the virus to diffuse away from the syringe before the syringe was retracted. For anatomical experiments, the Netrin-1 knockdown and scrambled control viruses were injected into the left and right hemispheres, with the type of virus injected into each hemisphere determined randomly. For behavioural experiments, the same virus was injected into both hemispheres.

## Behaviour – *Go/No-Go*

We used the Go/No-Go task to measure inhibitory control, as we have described previously (; *Reynolds et al., 2018*; *Reynolds et al., 2023*). The mice used for this experiment were adults (75±15 days of age at the beginning of the experiment) which had been stereotaxically injected with a Netrin-1 inhibiting or control virus at the onset of adolescence (see previous section).

During the experiment, mice were food restricted to 1.5 g food per to maintain a body weight of 85% of their initial free feeding weight. We used operant behavioural boxes (Med Associates, Inc, St Albans, VT, USA) equipped with a house light, an Sonalert tone generator, two illuminated nose-poke holes, and a pellet dispenser. Chocolate-flavoured dustless precision food pellets (BioServ, Inc, Flemington, NJ, USA) were used as our operant reinforcer. The experimental procedure consisted of two training stages, Discrimination Training and Reaction Time, followed by the Go/No-Go test phase. One session was conducted per mouse per day.

The first training stage is Discrimination Training. For this stage, at the start of each 20 min session, the house light comes on and remains illuminated throughout the session. Trials consist of the illumination of one nose-poke hole for 9 s, counterbalanced between nose-poke holes across mice. If the mouse does not nose-poke into the illuminated hole within that 9 s period, the cue light is extinguished for a 10 s inter-trial interval before the next trial. If the mouse responds to the cue light by nose-poking, they received a pellet and the trial is counted as a 'rewarded' trial. Responses to the active nose-poke hole when the cue light is off, as well as responses to the non-active nose-poke hole (where the cue light was never illuminated), were not rewarded. Mice received one Discrimination Training session per day until they reached a rate of 70% rewarded trials, at which point they advanced to the next stage of training.

The second training stage is Reaction Time. At this stage, mice were trained to respond only within 3 s of the cue illumination to receive the pellet reward. These training sessions lasted 30 min, but the house light does not remain illuminated throughout the session. Instead, the house light becomes illuminated for a variable amount of time (3, 6, or 9 s, distributed randomly) prior to the illumination of the cue light, to signal the start of a new trial. This is designed to signal for the mice to attend to the cue. If the mice responded during this pretrial period (a 'Premature Response'), the house light was turned off for a 10 s inter-trial interval and then a new trial is initiated. If the mouse did not perform a Premature Response, the cue light was illuminated for 3 s. A nose-poke into the illuminated hole during this 3 s period resulted in the delivery of a reward pellet. If the mouse did not respond, the cue and house lights were extinguished and a 10 s inter-trial interval was initiated, followed by a new trial. Mice received one Reaction Time training session per day until they reached a rate of 70% rewarded trials and fewer than 25% of trials ended due to a Premature Response, at which point they advanced to the Go/No-Go test stage.

After training, mice underwent 10 daily sessions of the Go/No-Go task. This task required the mice to respond to the illuminated cue light (a 'Go' trial) or to inhibit their response to this cue when it was presented in tandem with an 80 dB tone (a 'No-Go' trial) to receive a reward pellet. During a 'No-Go' trial, if mice responded during the 3 s presentation of both the illumination and tone cues, a 10 s inter-trial interval was initiated, followed by a new trial. A randomised, variable period of 3–9 s during which only the house light was illuminated signalled the start of each trial. A nose-poke during this time initiated a 10 s inter-trial period followed by a new trial. Within each session, the number of 'Go' and 'No-Go' trials were given in an approximately 1:1 ratio and presented in a randomised order. Each session lasted 30 min and consisted of approximately 60–100 completed trials.

We quantified three measures from the Go/No-Go test data. Commission errors were our measure of inhibitory control. A commission error occurs when a mouse nose-pokes during a 'No-Go' trial,

when the cue light is illuminated concurrently with the 80 dB tone. We also quantified omission errors, which are when a mouse fails to nose-poke during a 'Go' trial, when the cue light is illuminated in the absence of the tone. Finally, we calculated the correct response rate, which is the number of 'Go' trials where the mouse nose-pokes while the cue light is illuminated plus the number of 'No-Go' trials where the mouse does not nose-poke while the cue light is illuminated. All three measures are analysed as proportions of the total number of trials presented each test day.

### Behaviour – *light/dark box*

In hamsters, we used the light/dark box test, as we've described previously (*Kyne et al., 2019*). We used operant behavioural boxes consisting of two compartments: one with illumination from a house light (the light compartment; 40.0 cm × 39.9 cm × 31.2 cm) and one without illumination (the dark compartment; 38.9 cm × 12.7 cm × 15.2 cm). The compartments were separated by barrier with an opening that could be blocked by a metal door.

For each session, a hamster placed inside the dark compartment of the apparatus with the metal door closed. The session was initiated when the metal door was opened, allowing the hamster to explore the light compartment. The hamster was allowed to move freely between the two compartments for 10 min. We used the amount of time spent in the light compartment as our measure of exploratory behaviour.

The hamsters were recorded by a camera mounted above the boxes using Media Recorder 4 software (Noldus Information Technology Inc, Wageningen, The Netherlands). Scoring was done automatically using EthoVision XT10 software (Noldus Information Technology Inc, Wageningen, The Netherlands).

### Statistical analyses

Detailed statistical explanations for each analysis are presented in our Statistics Supplement. All analyses were conducted in the statistical programming language R (*R Development Core Team, 2014*). For all analyses our significance threshold was set at $p=0.05$.

## Acknowledgements

We are grateful for the technical assistance of the Pathology Core, Centre for Phenogenomics, Toronto, Canada, and in particular Milan Ganguly and Gregory Ossetchkine, for their excellent histological assistance. Furthermore, we are grateful to Helen Cooper and Philip Vassilev for their critical and insightful readings of our manuscript. Funding: National Institute on Drug Abuse grant R01DA037911 (CF). Canadian Institutes of Health Research FRN: 156272 (CF). Canadian Institutes of Health Research FRN: 170130 (CF). National Science and Engineering Research Council of Canada RGPIN-2020-04703 (CF). National Science Foundation IOS-1754878 (MJP). National Science and Engineering Research Council of Canada PGSD3-415253-2012 (DH). Quebec Nature and Technology Research Fund 208332 (DH). National Science and Engineering Research Council of Canada PDF5171462018 (DH). McGill-Douglas Max Planck Institute of Psychiatry International Collaborative Initiative in Adversity and Mental Health, an international partnership funded by the Canada First Research Excellence Fund, awarded to McGill University for the Healthy Brains for Healthy Lives initiative (RGA).

## Additional information

### Funding

| Funder | Grant reference number | Author |
| --- | --- | --- |
| National Institute on Drug Abuse | R01DA037911 | Cecilia Flores |
| Canadian Institutes of Health Research | FRN: 156272 | Cecilia Flores |
| Canadian Institutes of Health Research | FRN: 170130 | Cecilia Flores |

| Funder | Grant reference number | Author |
| --- | --- | --- |
| National Science and Engineering Research Council of Canada | RGPIN-2020-04703 | Cecilia Flores |
| National Science Foundation | IOS-1754878 | Matthew J Paul |

The funders had no role in study design, data collection and interpretation, or the decision to submit the work for publication.

### Author contributions

Daniel Hoops, Conceptualization, Data curation, Formal analysis, Supervision, Validation, Investigation, Visualization, Methodology, Writing - original draft, Writing – review and editing; Robert Kyne, Samer Salameh, Elise Ewing, Alina Tao He, Taylor Orsini, Anais Durand, Christina Popescu, Janet Mengyi Zhao, Kelcie Shatz, LiPing Li, Quinn Carroll, Investigation; Del MacGowan, Radu Gabriel Avramescu, Investigation, Visualization; Guofa Liu, Resources; Matthew J Paul, Cecilia Flores, Conceptualization, Resources, Supervision, Funding acquisition, Methodology, Writing – review and editing

### Author ORCIDs

Daniel Hoops http://orcid.org/0000-0002-5707-3513
Anais Durand https://orcid.org/0000-0001-8125-1394
Matthew J Paul http://orcid.org/0000-0002-9097-1390
Cecilia Flores https://orcid.org/0000-0001-8606-5910

Reviewer #1 (Public Review): https://doi.org/10.7554/eLife.88261.4.sa1
Reviewer #2 (Public Review): https://doi.org/10.7554/eLife.88261.4.sa2
Reviewer #3 (Public Review): https://doi.org/10.7554/eLife.88261.4.sa3
Author response https://doi.org/10.7554/eLife.88261.4.sa4

## Additional files

### Supplementary files

• MDAR checklist

### Data availability

All data and code use in these analyses are available through the Open Science Framework.

The following dataset was generated:

| Author(s) | Year | Dataset title | Dataset URL | Database and Identifier |
| --- | --- | --- | --- | --- |
| Flores et al. | 2024 | The scheduling of adolescence with Netrin-1 and UNC5C | https://doi.org/10.17605/OSF.IO/DU3H4 | Open Science Framework, 10.17605/OSF.IO/DU3H4 |

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

## Appendix 1

### Supplementary analyses
Supplementary analysis 1

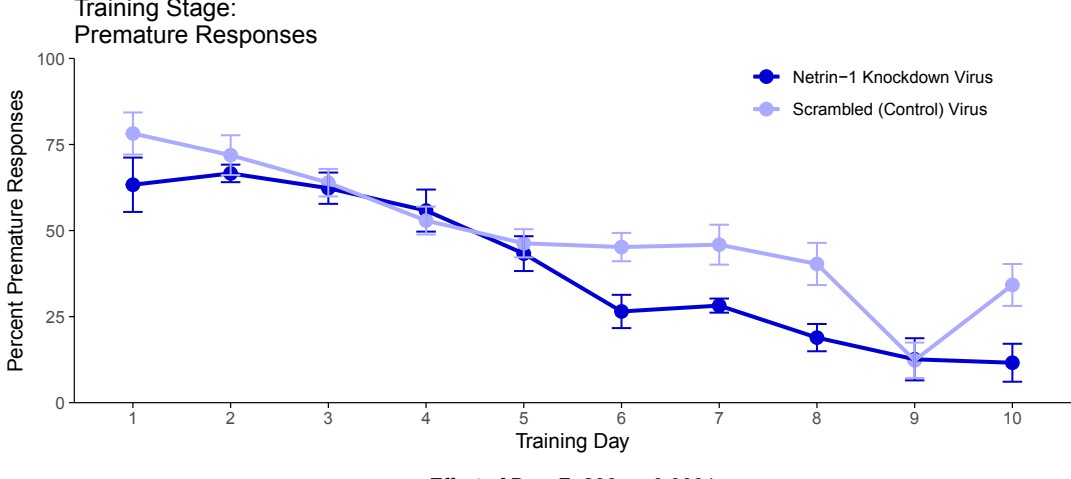

Effect of Day: F=236, p<0.0001
Virus/Day Interaction: F=2.25, p=0.13

**Appendix 1—figure 1.** This graph illustrates the ability of adult (60+ days of age) mice to perform a learned response to a stimulus in order to receive a food reward as part of a learning phase leading up to the Go/No-Go behavioural experiment. 'Percent Premature Responses' quantifies the percentage of trials in which the mice performed the learned response before the onset of the cue, and therefore ended the trial without receiving a reward. The statistical results presented below the graph are from the analysis of variance (ANOVA) presented below. The data presented in this graph and analyzed below correspond to the second training stage of the Go/No-Go behavioural experiment, referred to as Reaction Time. See our Methods subsection 'Behaviour – Go/No-Go' for details.

In this experiment, at the onset of adolescence (21 days of age) the mice received an injection of a virus that either knocked down Netrin-1 or served as a nonfunctional control. The viruses were injected along the route by which dopamine axons grow from the nucleus accumbens to the medial prefrontal cortex during adolescence. Behavioural training and testing then commenced in adulthood (75 ± 15 days of age). More results from this Go/No-Go experiment are presented in *Figure 1* of our paper.

## Summary statistics

| Virus | Day | N | Percent premature | sd | se | ci |
|---|---|---|---|---|---|---|
| Netrin-1 knockdown virus | 1 | 10 | 0.63 | 0.25 | 0.08 | 0.18 |
| Netrin-1 knockdown virus | 2 | 10 | 0.67 | 0.08 | 0.03 | 0.06 |
| Netrin-1 knockdown virus | 3 | 10 | 0.62 | 0.14 | 0.05 | 0.1 |
| Netrin-1 knockdown virus | 4 | 10 | 0.56 | 0.19 | 0.06 | 0.14 |
| Netrin-1 knockdown virus | 5 | 10 | 0.43 | 0.16 | 0.05 | 0.11 |
| Netrin-1 knockdown virus | 6 | 10 | 0.26 | 0.15 | 0.05 | 0.11 |
| Netrin-1 knockdown virus | 7 | 10 | 0.28 | 0.06 | 0.02 | 0.05 |
| Netrin-1 knockdown virus | 8 | 10 | 0.19 | 0.13 | 0.04 | 0.09 |
| Netrin-1 knockdown virus | 9 | 10 | 0.13 | 0.19 | 0.06 | 0.14 |
| Netrin-1 knockdown virus | 10 | 10 | 0.12 | 0.17 | 0.06 | 0.12 |

*Continued on next page*

*Continued*

| Virus | Day | N | Percent premature | sd | se | ci |
|---|---|---|---|---|---|---|
| Scrambled (control) virus | 1 | 10 | 0.78 | 0.19 | 0.06 | 0.14 |
| Scrambled (control) virus | 2 | 10 | 0.72 | 0.18 | 0.06 | 0.13 |
| Scrambled (control) virus | 3 | 10 | 0.64 | 0.13 | 0.04 | 0.09 |
| Scrambled (control) virus | 4 | 10 | 0.53 | 0.13 | 0.04 | 0.09 |
| Scrambled (control) virus | 5 | 10 | 0.46 | 0.13 | 0.04 | 0.09 |
| Scrambled (control) virus | 6 | 10 | 0.45 | 0.13 | 0.04 | 0.09 |
| Scrambled (control) virus | 7 | 10 | 0.46 | 0.18 | 0.06 | 0.13 |
| Scrambled (control) virus | 8 | 10 | 0.4 | 0.19 | 0.06 | 0.14 |
| Scrambled (control) virus | 9 | 10 | 0.12 | 0.16 | 0.05 | 0.12 |
| Scrambled (control) virus | 10 | 10 | 0.34 | 0.19 | 0.06 | 0.14 |

An analysis of variance (ANOVA) revealed no significant differences between viral treatments in the percent of responses that were premature. We determined this using a mixed-effects ANOVA with virus and day as fixed effects, mouse ID as a random effect, and percent of premature responses as the response variable.

## Model output

| Term | Statistic | df | p-Value |
|---|---|---|---|
| Day | 235.750804 | 1 | 0.0000000 |
| Virus | 2.167892 | 1 | 0.1409192 |
| Day:virus | 2.253370 | 1 | 0.1333237 |

## Supplementary analysis 2

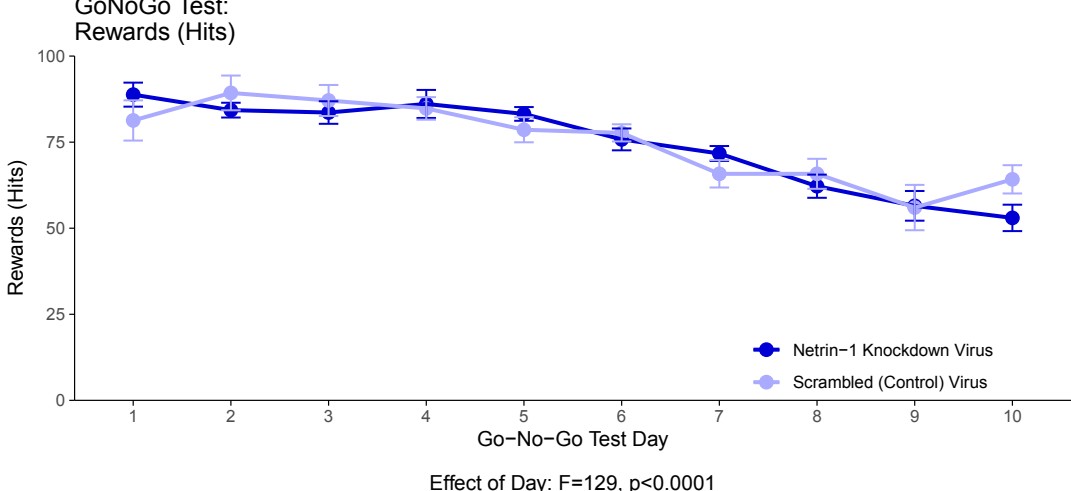

Effect of Day: F=129, p<0.0001
Virus/Day Interaction: F=0.77, p=0.38

**Appendix 1—figure 2.** This graph illustrates the ability of adult (60+ days of age) mice to perform a learned response to a stimulus in order to receive a food reward as part of the Go/No-Go behavioural experiment. 'Hits' quantifies the percentage of trials in which the mice performed correctly in response to a visual cue and in the absence of an auditory cue (a 'hit'), and received a food reward as a result. The statistical results presented below the graph are from the analysis of variance (ANOVA) presented below. The data presented in this graph and analyzed below correspond to the test stage of the Go/No-Go behavioural experiment, referred to as the Go/No-Go task. See our Methods subsection 'Behaviour – Go/No-Go' for details.

In this experiment, at the onset of adolescence (21 days of age) the mice received an injection of a virus that either knocked down Netrin-1 or served as a nonfunctional control. The viruses were injected along the route by which dopamine axons grow from the nucleus accumbens to the mPFC during adolescence. Behavioural training and testing then commenced in adulthood (75 ± 15 days of age). More results from this Go/No-Go experiment are presented in *Figure 1* of our paper.

## Summary statistics

| Virus | Day | N | Hits | sd | se | ci |
|---|---|---|---|---|---|---|
| Netrin-1 knockdown virus | 1 | 10 | 0.89 | 0.11 | 0.03 | 0.08 |
| Netrin-1 knockdown virus | 2 | 10 | 0.84 | 0.07 | 0.02 | 0.05 |
| Netrin-1 knockdown virus | 3 | 10 | 0.84 | 0.10 | 0.03 | 0.07 |
| Netrin-1 knockdown virus | 4 | 10 | 0.86 | 0.13 | 0.04 | 0.09 |
| Netrin-1 knockdown virus | 5 | 10 | 0.83 | 0.06 | 0.02 | 0.05 |
| Netrin-1 knockdown virus | 6 | 10 | 0.76 | 0.10 | 0.03 | 0.07 |
| Netrin-1 knockdown virus | 7 | 10 | 0.72 | 0.07 | 0.02 | 0.05 |
| Netrin-1 knockdown virus | 8 | 10 | 0.62 | 0.11 | 0.03 | 0.08 |
| Netrin-1 knockdown virus | 9 | 10 | 0.56 | 0.14 | 0.04 | 0.10 |
| Netrin-1 knockdown virus | 10 | 8 | 0.53 | 0.11 | 0.04 | 0.09 |
| Scrambled (control) virus | 1 | 10 | 0.81 | 0.18 | 0.06 | 0.13 |
| Scrambled (control) virus | 2 | 10 | 0.89 | 0.16 | 0.05 | 0.11 |
| Scrambled (control) virus | 3 | 10 | 0.87 | 0.14 | 0.05 | 0.10 |
| Scrambled (control) virus | 4 | 10 | 0.85 | 0.10 | 0.03 | 0.07 |
| Scrambled (control) virus | 5 | 10 | 0.79 | 0.12 | 0.04 | 0.08 |
| Scrambled (control) virus | 6 | 10 | 0.78 | 0.08 | 0.02 | 0.06 |
| Scrambled (control) virus | 7 | 10 | 0.66 | 0.13 | 0.04 | 0.09 |
| Scrambled (control) virus | 8 | 10 | 0.66 | 0.14 | 0.04 | 0.10 |
| Scrambled (control) virus | 9 | 10 | 0.56 | 0.21 | 0.07 | 0.15 |
| Scrambled (control) virus | 10 | 10 | 0.64 | 0.13 | 0.04 | 0.09 |

An analysis of variance (ANOVA) revealed no significant differences between viral treatments in the ability of the mice to respond correctly to a visual cue. We determined this using a mixed-effects ANOVA with virus and day as fixed effects, mouse ID as a random effect, and percent of responses that were hits as the response variable.

## Model output

| Term | Statistic | df | p-Value |
|---|---|---|---|
| Day | 128.8139852 | 1 | 0.0000000 |
| Virus | 0.0006105 | 1 | 0.9802884 |
| Day:virus | 0.7673066 | 1 | 0.3810516 |

## Supplementary analysis 3

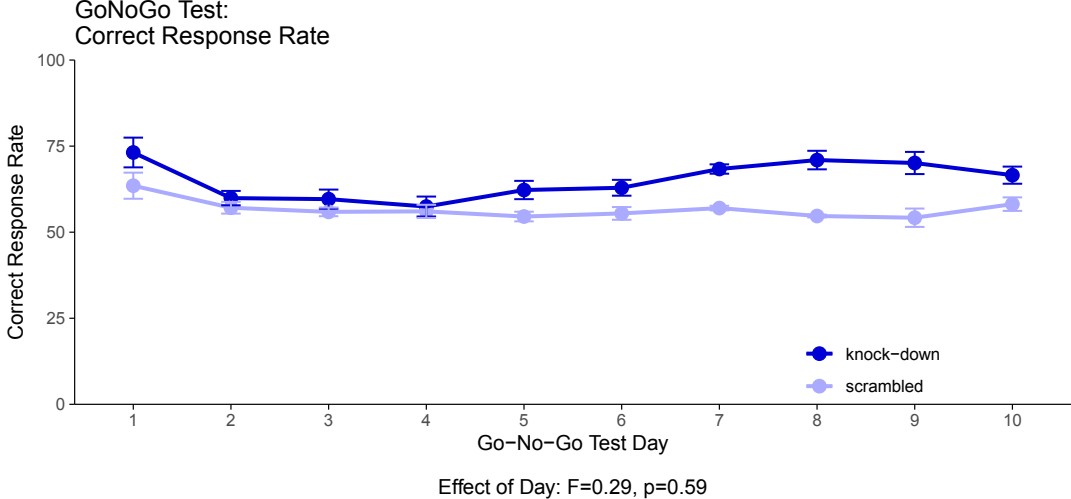

**Appendix 1—figure 3.** This graph illustrates the ability of adult (60+ day old) mice to perform both 'Go' trials and 'No-Go' trials correctly. A 'Go' trial requires a behavioural response to a visual cue. A 'No-Go' trial requires the inhibition of the behavioural response to the visual cue when it is presented with a second, auditory cue. The 'Correct Response Rate' quantifies the trials where the mice respond correctly whether the trial is a 'Go' trial or a 'No-Go' trial. The statistical results presented below the graph are from the analysis of variance (ANOVA) presented below. The data presented in this graph and analysed below correspond to the test stage of the Go/No-Go behavioural experiment, referred to as the Go/No-Go task. See our Methods subsection 'Behavior – Go/No-Go' for details.

In this experiment, at the onset of adolescence (21 days of age) the mice received an injection of a virus that either knocked down Netrin-1 or served as a nonfunctional control. The viruses were injected along the route by which dopamine axons grow from the nucleus accumbens to the medial prefrontal cortex during adolescence. Behavioural training and testing then commenced in adulthood (75 ± 15 days of age). More results from this Go/No-Go experiment are presented in *Figure 1* of our paper.

## Summary statistics

| Virus | Day | N | Correct response rate | sd | se | ci |
|---|---|---|---|---|---|---|
| Netrin-1 knockdown virus | 1 | 10 | 0.73 | 0.14 | 0.04 | 0.10 |
| Netrin-1 knockdown virus | 2 | 10 | 0.60 | 0.07 | 0.02 | 0.05 |
| Netrin-1 knockdown virus | 3 | 10 | 0.60 | 0.09 | 0.03 | 0.06 |
| Netrin-1 knockdown virus | 4 | 10 | 0.57 | 0.09 | 0.03 | 0.07 |
| Netrin-1 knockdown virus | 5 | 10 | 0.62 | 0.08 | 0.03 | 0.06 |
| Netrin-1 knockdown virus | 6 | 10 | 0.63 | 0.07 | 0.02 | 0.05 |
| Netrin-1 knockdown virus | 7 | 10 | 0.68 | 0.04 | 0.01 | 0.03 |
| Netrin-1 knockdown virus | 8 | 10 | 0.71 | 0.09 | 0.03 | 0.06 |
| Netrin-1 knockdown virus | 9 | 10 | 0.70 | 0.10 | 0.03 | 0.07 |
| Netrin-1 knockdown virus | 10 | 8 | 0.67 | 0.07 | 0.02 | 0.06 |
| Scrambled (control) virus | 1 | 10 | 0.64 | 0.12 | 0.04 | 0.09 |
| Scrambled (control) virus | 2 | 10 | 0.57 | 0.05 | 0.02 | 0.04 |

*Continued on next page*

*Continued*

| Virus | Day | N | Correct response rate | sd | se | ci |
|---|---|---|---|---|---|---|
| Scrambled (control) virus | 3 | 10 | 0.56 | 0.04 | 0.01 | 0.03 |
| Scrambled (control) virus | 4 | 10 | 0.56 | 0.06 | 0.02 | 0.04 |
| Scrambled (control) virus | 5 | 10 | 0.55 | 0.04 | 0.01 | 0.03 |
| Scrambled (control) virus | 6 | 10 | 0.55 | 0.06 | 0.02 | 0.04 |
| Scrambled (control) virus | 7 | 10 | 0.57 | 0.02 | 0.01 | 0.02 |
| Scrambled (control) virus | 8 | 10 | 0.55 | 0.01 | 0.00 | 0.01 |
| Scrambled (control) virus | 9 | 10 | 0.54 | 0.08 | 0.03 | 0.06 |
| Scrambled (control) virus | 10 | 10 | 0.58 | 0.06 | 0.02 | 0.04 |

An analysis of variance (ANOVA) revealed a significant difference between viral treatments in the ability of the mice to respond correctly to cues during the Go/No-Go test. We determined this using a mixed-effects ANOVA with virus and day as fixed effects, mouse ID as a random effect, and percent of responses that were correct as the response variable.

## Model output

| Term | Statistic | df | p-Value |
|---|---|---|---|
| Day | 0.2906515 | 1 | 0.5898033 |
| Virus | 8.3375188 | 1 | 0.0038835 |
| Day:virus | 7.2103291 | 1 | 0.0072485 |

## Supplementary analysis 4

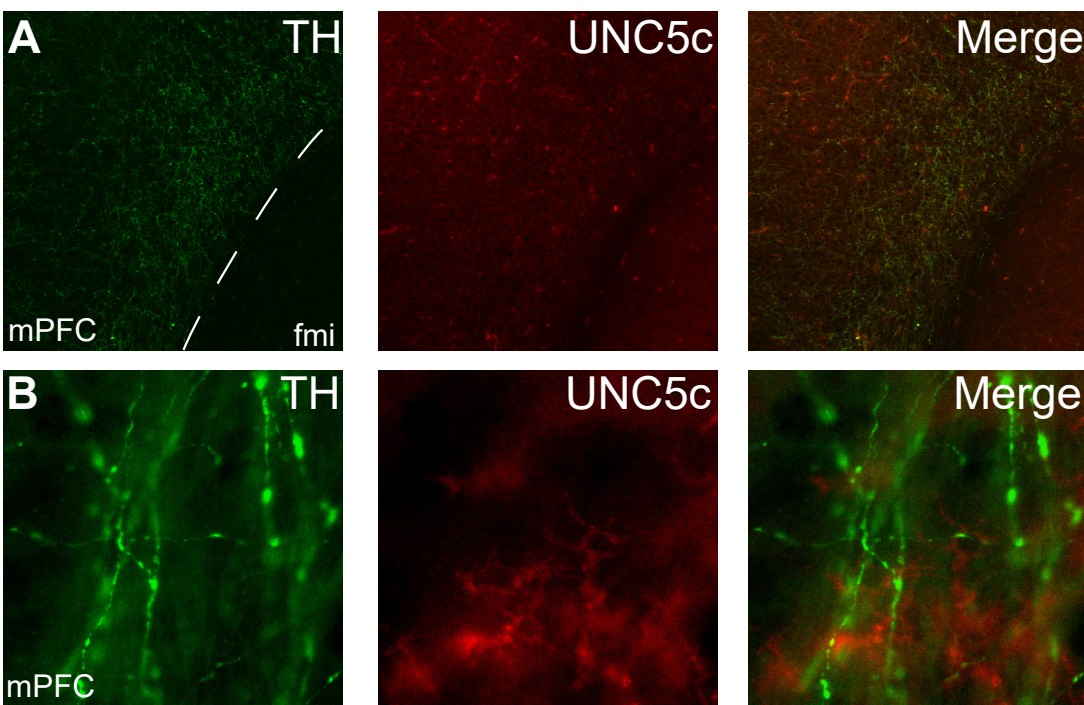

**Appendix 1—figure 4.** Expression of UNC5c protein in the medial prefrontal cortex of an adult male mouse. Low (**A**) and high (**B**) magnification images demonstrate that there is little UNC5c expression in dopamine axons in the medial prefrontal cortex. Here, we identify dopamine axons by immunofluorescent staining for tyrosine
*Appendix 1—figure 4 continued on next page*

*Appendix 1—figure 4 continued*
hydroxylase (TH). See our Methods subsection 'Immunohistochemistry' for details. Abbreviations: fmi: forceps minor of the corpus callosum, mPFC: medial prefrontal cortex.

## Supplementary analysis 5

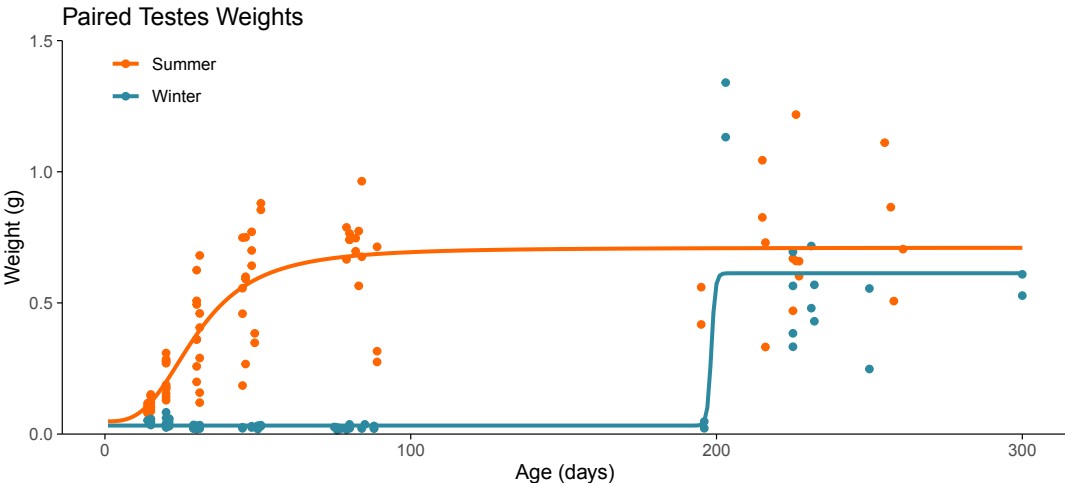

Effect of Daylength: F=−10.13, p<0.0001

**Appendix 1—figure 5.** This graph illustrates the effect of daylength on the reproductive development of male hamsters. Testicular weight is a commonly used proxy for the timing of puberty in male hamsters. The increase in paired testes weight, signalling puberty, is delayed when the hamsters are housed under a winter-mimicking short daylength, compared to a summer-mimicking long daylength. The statistical result below the graph is from the linear model presented below. For details on the hamster treatment protocol, see our Methods subsection 'Animals'.

## Summary statistics

| Photo | N | Gonad weight | sd | se | ci |
|---|---|---|---|---|---|
| Summer | 83 | 0.4632289 | 0.2858927 | 0.0313808 | 0.0624264 |
| Winter | 73 | 0.1442466 | 0.2626273 | 0.0307382 | 0.0612755 |

A linear regression revealed a significant difference between daylength treatments in testes weight. We determined this using a linear model with daylength and age as fixed effects, and paired testes weight as the response variable. More results from male hamsters raised under summer- and winter-mimicking daylengths are presented in *Figure 3* of our paper.

## Model output

| Term | Estimate | Std. error | Statistic | p-Value |
|---|---|---|---|---|
| (Intercept) | 0.2969836 | 0.0317056 | 9.3669101 | 0.0000000 |
| PhotoWinter | −0.3492544 | 0.0464937 | −7.5118722 | 0.0000000 |
| Age.Test | 0.0021778 | 0.0002912 | 7.4786232 | 0.0000000 |
| PhotoWinter:Age.Test | 0.0001891 | 0.0004102 | 0.4610736 | 0.6454049 |

## Supplementary analysis 6

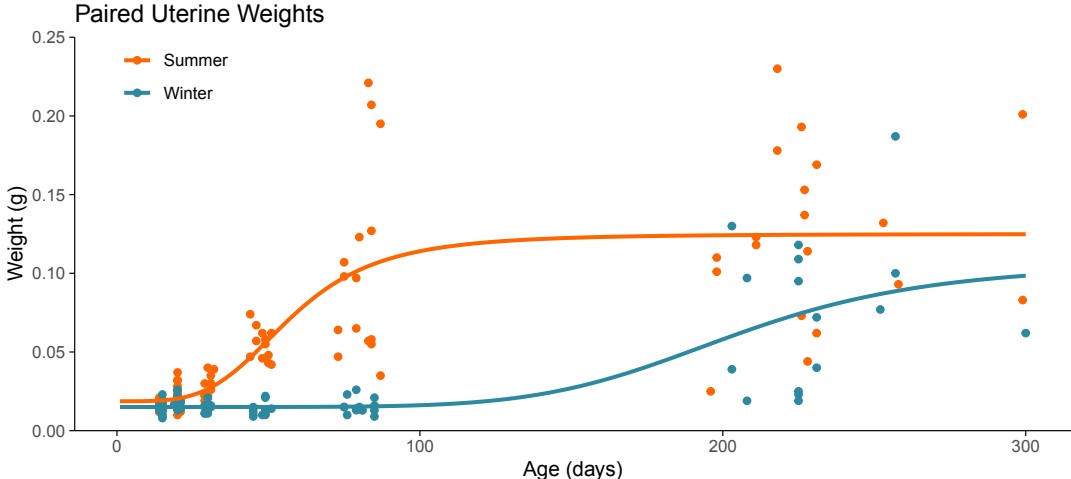

**Appendix 1—figure 6.** This graph illustrates the effect of daylength on the reproductive development of female hamsters. Uterine weight is a commonly used proxy for the timing of puberty in female hamsters. The increase in uterine weight, signalling puberty, is delayed in hamsters that are housed under a winter-mimicking short daylength, compared to a summer-mimicking long daylength. The statistical result below the graph is from the linear model presented below. For details on the hamster treatment protocol, see our Methods subsection 'Animals'.

## Summary statistics

| Photo | N | Gonad weight | sd | se | ci |
|---|---|---|---|---|---|
| Summer | 85 | 0.0636471 | 0.0566068 | 0.0061399 | 0.0122098 |
| Winter | 77 | 0.0277013 | 0.0328422 | 0.0037427 | 0.0074543 |

A linear regression revealed a significant difference between daylength treatments in uterine weight. We determined this using a linear model with daylength and age as fixed effects, and paired uterine weight as the response variable. More results from female hamsters raised under summer- and winter-mimicking daylengths are presented in *Figure 4* of our paper.

## Model output

| Term | Estimate | Std. error | Statistic | p-Value |
|---|---|---|---|---|
| (Intercept) | 0.0252757 | 0.0051360 | 4.921270 | 0.0000021 |
| PhotoWinter | −0.0210310 | 0.0074302 | −2.830476 | 0.0052522 |
| Age.Test | 0.0004647 | 0.0000437 | 10.623410 | 0.0000000 |
| PhotoWinter:Age.Test | −0.0001704 | 0.0000643 | −2.649543 | 0.0088792 |

## Supplementary analysis 7

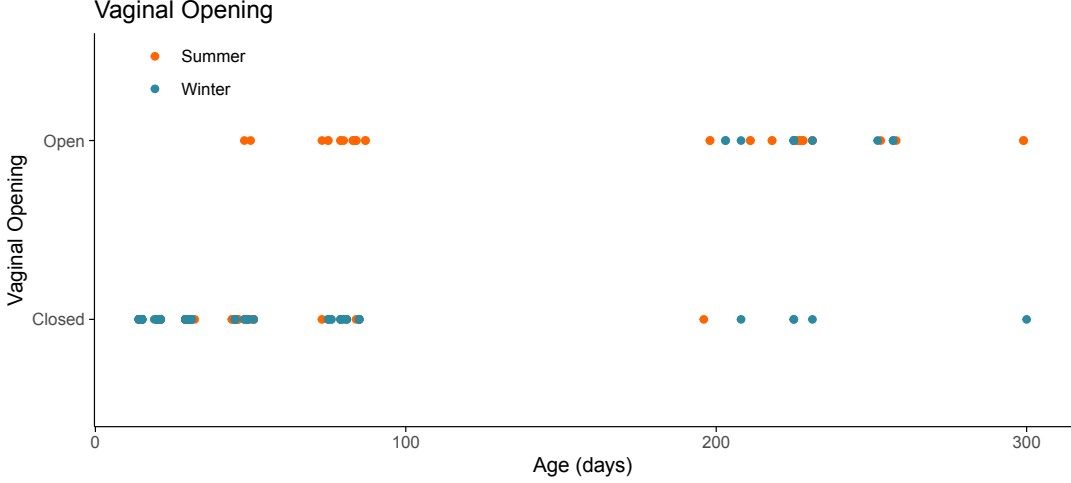

**Appendix 1—figure 7.** This graph illustrates the effects of daylength on the reproductive development of female hamsters. The timing of the opening of the vagina is, in females, a commonly used proxy for the timing of puberty alongside uterine weight. The opening of the vagina, signalling puberty, is delayed when housed under a winter-mimicking short daylength, compared to a summer-mimicking long daylength. The statistical result below the graph is from the model presented below. For details on the hamster treatment protocol, see our Methods subsection 'Animals'.

### Summary statistics

| Photo | Vaginal opening | N |
| --- | --- | --- |
| Summer | Closed | 52 |
| Winter | Closed | 66 |
| Summer | Open | 33 |
| Winter | Open | 11 |

A generalised linear regression revealed a significant difference between daylength treatments in the timing of vaginal opening. We determined this using a generalised linear model with daylength and age as fixed effects, and vaginal openness (dichotomous, 0=closed and 1=open) as the response variable. More results from female hamsters raised under summer- and winter-mimicking daylengths are presented in *Figure 4* of our paper.

### Model output

| Term | Estimate | Std. error | Statistic | p-Value |
| --- | --- | --- | --- | --- |
| (Intercept) | 0.0239557 | 0.0435168 | 0.5504926 | 0.5827590 |
| PhotoWinter | –0.1364297 | 0.0629550 | –2.1670980 | 0.0317252 |
| Age.Test | 0.0044114 | 0.0003706 | 11.9031505 | 0.0000000 |
| PhotoWinter:Age.Test | –0.0012078 | 0.0005448 | –2.2169133 | 0.0280570 |

## Supplementary analysis 8

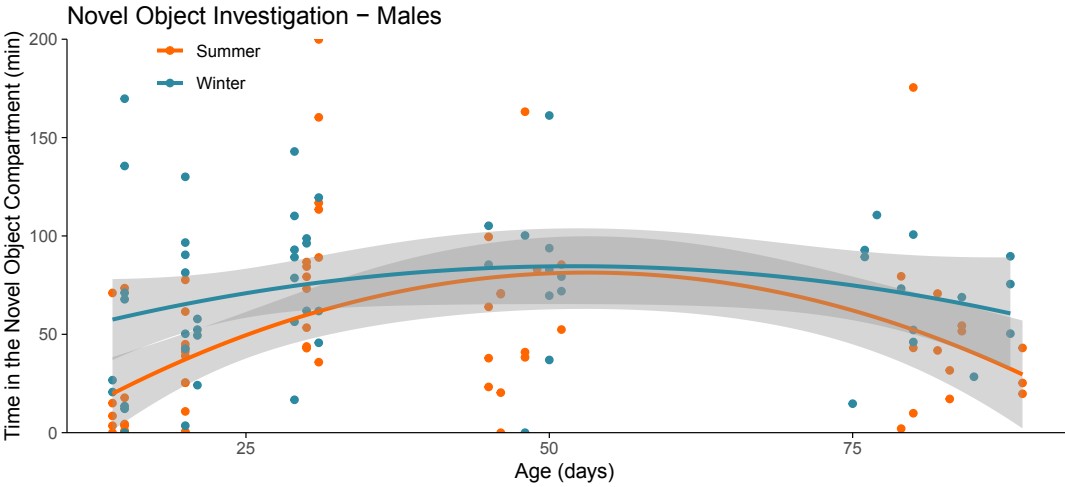

**Appendix 1—figure 8.** This graph illustrates the results of a novel object behavioural experiment with male summer and winter hamsters between the ages of 15 and 90 days. The protocol used for this experiment was identical to that described in the Methods subsection 'Behaviour – light/dark box', the only difference being that for this experiment a novel object (a small wire-framed rodent cage) was placed in the light compartment of the box. The time spent in the compartment of the box that contained the novel object was quantified as our measure of novel object investigation. Summer hamsters were housed under a summer-mimicking long photoperiod, while winter hamsters were housed under a winter-mimicking short photoperiod. For more details on our hamster housing protocol, see our Methods subsection 'Animals'. The statistical result below the graph is from the polynomial model shown below.

## Summary statistics

| Photo | N | NovObj | sd | se | ci |
|---|---|---|---|---|---|
| Summer | 66 | 49.42719 | 44.92317 | 5.529660 | 11.04349 |
| Winter | 57 | 70.98883 | 39.29375 | 5.204584 | 10.42603 |

A polynomial regression shows a significant effect of daylength on novel object investigation in male hamsters. We found this using a fourth-order polynomial regression with age as the polynomial predictor variable, daylength as the categorical predictor variable, and time in the compartment containing the novel object as the response variable. More results from male hamsters raised under summer- and winter-mimicking daylengths are presented in *Figure 3* of our paper.

## Model output

| Term | Estimate | Std. error | Statistic | p-Value |
|---|---|---|---|---|
| (Intercept) | 50.74917 | 4.882881 | 10.3932842 | 0.0000000 |
| poly(Age.Test, 4)1 | 61.66611 | 53.996207 | 1.1420451 | 0.2558494 |
| poly(Age.Test, 4)2 | –204.24463 | 52.908097 | –3.8603662 | 0.0001890 |
| poly(Age.Test, 4)3 | 99.91338 | 57.107338 | 1.7495717 | 0.0829071 |
| poly(Age.Test, 4)4 | –86.95956 | 54.573615 | –1.5934360 | 0.1138558 |
| PhotoWinter | 19.90223 | 7.179115 | 2.7722397 | 0.0065130 |
| poly(Age.Test, 4)1:PhotoWinter | –33.55631 | 79.965778 | –0.4196333 | 0.6755501 |

*Continued on next page*

*Continued*

| Term | Estimate | Std. error | Statistic | p-Value |
|------|----------|-----------|-----------|---------|
| poly(Age.Test, 4)2:PhotoWinter | 104.27374 | 80.642472 | 1.2930374 | 0.1986355 |
| poly(Age.Test, 4)3:PhotoWinter | –65.26912 | 79.942686 | –0.8164489 | 0.4159613 |
| poly(Age.Test, 4)4:PhotoWinter | 75.71783 | 79.844049 | 0.9483216 | 0.3449894 |

## Supplementary analysis 9

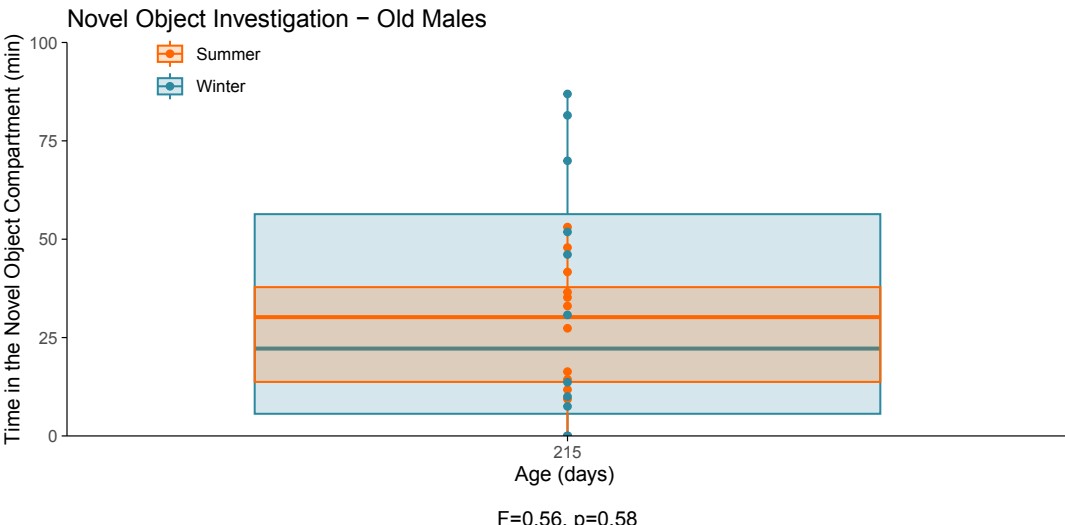

F=0.56, p=0.58

**Appendix 1—figure 9.** This graph illustrates the results of a novel object behavioural experiment with 215-day-old male summer and winter hamsters. The protocol used for this experiment was identical to that described in the Methods subsection 'Behaviour – light/dark box', the only difference being that for this experiment a novel object (a small wire-framed rodent cage) was placed in the light compartment of the box. The time spent in the compartment of the box that contained the novel object was quantified as our measure of novel object investigation. Summer hamsters were housed under a summer-mimicking long photoperiod, while winter hamsters were housed under a winter-mimicking short photoperiod. For more details on our hamster housing protocol, see our Methods subsection 'Animals'. The statistical result below the graph is from the linear model shown below.

### Summary statistics

| Photo | N | NovObj | sd | se | ci |
|-------|---|--------|-----|-----|-----|
| Summer | 12 | 27.21830 | 16.70891 | 4.823448 | 10.61634 |
| Winter | 12 | 33.18527 | 32.99246 | 9.524102 | 20.96241 |

A linear regression showed no difference in novel object investigation between the old male summer and winter hamsters. We found this using a linear regression with daylength as the predictor variable and time in the compartment with the novel object as the response variable. More results from male hamsters raised under summer- and winter-mimicking daylengths are presented in *Figure 3* of our paper.

### Model output

| Term | Estimate | Std. error | Statistic | p-Value |
|------|----------|-----------|-----------|---------|
| (Intercept) | 27.218298 | 7.548979 | 3.6055603 | 0.0015703 |
| PhotoWinter | 5.966972 | 10.675868 | 0.5589215 | 0.5818606 |

## Supplementary analysis 10

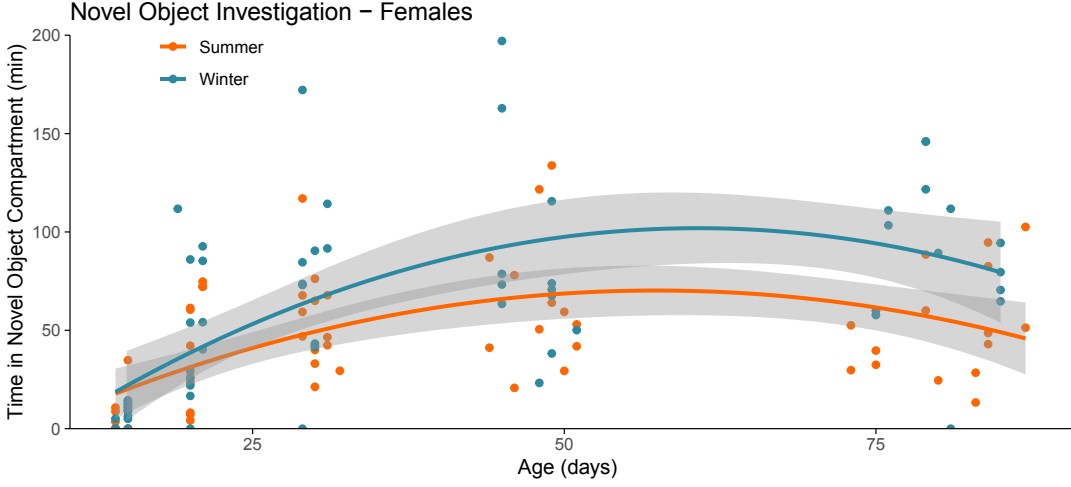

**Appendix 1—figure 10.** This graph illustrates the results of a novel object behavioural experiment with female summer and winter hamsters between the ages of 15 and 90 days. The protocol used for this experiment was identical to that described in the Methods subsection 'Behaviour – light/dark box", the only difference being that for this experiment a novel object (a small wire-framed rodent cage) was placed in the light compartment of the box. The time spent in the compartment of the box that contained the novel object was quantified as our measure of novel object investigation. Summer hamsters were housed under a summer-mimicking long photoperiod, while winter hamsters were housed under a winter-mimicking short photoperiod. For more details on our hamster housing protocol, see our Methods subsection 'Animals'. The statistical result below the graph is from the polynomial model shown below.

## Summary statistics

| Photo | N | NovObj | sd | se | ci |
|-------|----|----------|----------|----------|-----------|
| Summer | 66 | 44.69326 | 31.92605 | 3.929825 | 7.848403 |
| Winter | 61 | 62.38676 | 47.92103 | 6.135659 | 12.273145 |

A polynomial regression shows a significant effect of daylength on novel object investigation in female hamsters. We found this using a fourth-order polynomial regression with age as the polynomial predictor variable, daylength as the categorical predictor variable, and time in the compartment containing the novel object as the response variable. More results from female hamsters raised under summer- and winter-mimicking daylengths are presented in *Figure 4* of our paper.

## Model output

| Term | Estimate | Std. error | Statistic | p-Value |
|------|----------|------------|-----------|---------|
| (Intercept) | 44.350231 | 3.981734 | 11.1384199 | 0.0000000 |
| poly(Age.Test, 4)1 | 125.375406 | 44.902631 | 2.7921617 | 0.0061174 |
| poly(Age.Test, 4)2 | −144.834167 | 44.552894 | −3.2508363 | 0.0015031 |
| poly(Age.Test, 4)3 | 108.070413 | 43.036831 | 2.5111146 | 0.0133996 |
| poly(Age.Test, 4)4 | 9.230251 | 44.065269 | 0.2094677 | 0.8344474 |
| PhotoWinter | 17.957961 | 5.748067 | 3.1241737 | 0.0022482 |
| poly(Age.Test, 4)1:PhotoWinter | 122.752522 | 65.140090 | 1.8844389 | 0.0619871 |

*Continued on next page*

*Continued*

| Term | Estimate | Std. error | Statistic | p-Value |
|---|---|---|---|---|
| poly(Age.Test, 4)2:PhotoWinter | −47.920292 | 65.137957 | −0.7356739 | 0.4634015 |
| poly(Age.Test, 4)3:PhotoWinter | −14.910892 | 65.638856 | −0.2271656 | 0.8206914 |
| poly(Age.Test, 4)4:PhotoWinter | −100.220550 | 65.612433 | −1.5274628 | 0.1293449 |

## Supplementary analysis 11

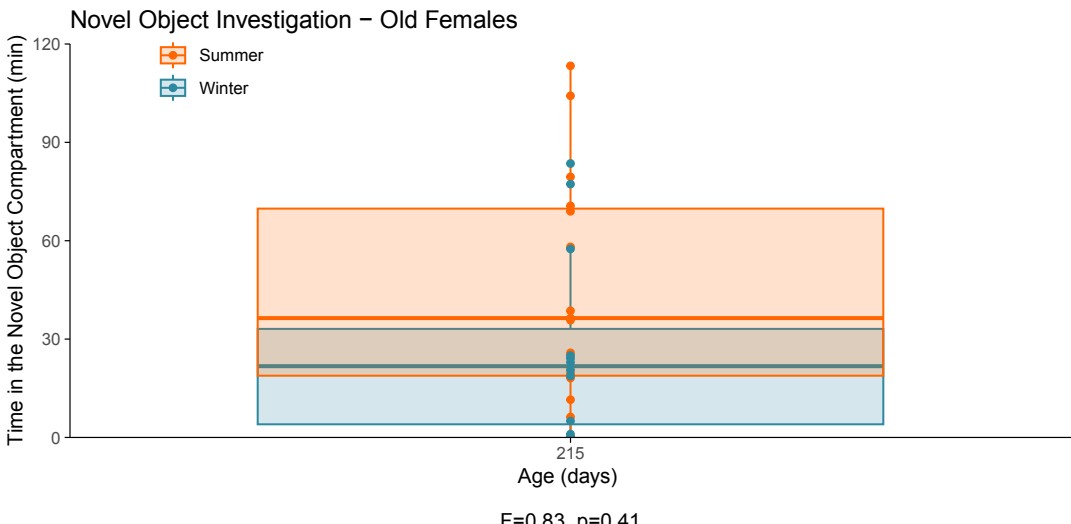

F=0.83, p=0.41

**Appendix 1—figure 11.** This graph illustrates the results of a novel object behavioural experiment with 215-day-old female summer and winter hamsters. The protocol used for this experiment was identical to that described in the Methods subsection 'Behaviour – light/dark box', the only difference being that for this experiment a novel object (a small wire-framed rodent cage) was placed in the light compartment of the box. The time spent in the compartment of the box that contained the novel object was quantified as our measure of novel object investigation. Summer hamsters were housed under a summer-mimicking long photoperiod, while winter hamsters were housed under a winter-mimicking short photoperiod. For more details on our hamster housing protocol, see our Methods subsection 'Animals'. The statistical result below the graph is from the linear model shown below.

## Summary statistics

| Photo | N | NovObj | sd | se | ci |
|---|---|---|---|---|---|
| Summer | 27 | 37.53426 | 29.55776 | 5.688393 | 11.69266 |
| Winter | 24 | 30.57994 | 30.59866 | 6.245926 | 12.92068 |

A linear regression showed no difference in novel object investigation between the old female summer and winter hamsters. We found this using a linear regression with daylength as the predictor variable and time in the compartment with the novel object as the response variable. More results from female hamsters raised under summer- and winter-mimicking daylengths are presented in *Figure 4* of our paper.

## Model output

| Term | Estimate | Std. error | Statistic | p-Value |
|---|---|---|---|---|
| (Intercept) | 37.534260 | 5.783286 | 6.4901269 | 0.0000000 |
| PhotoWinter | −6.954318 | 8.430516 | −0.8248983 | 0.4134271 |

