## [Editor Report · eLife assessment]

This study addresses an **important**, understudied question using approaches that link molecular, circuit, and behavioral changes. The findings that Netrin-1 and UNC5c can guide dopaminergic innervation from the nucleus accumbens to the cortex during adolescence are **solid**. The data showing that the onset of Unc5 expression is sexually dimorphic in mice, and that in Siberian hamsters environmental effects on development are also sexually dimorphic are also **solid**. Reviewers identified significant gaps in evidence for specificity of Netrin-1 expression, which, if filled, would strengthen the evidence for some of the claims. Future work would also benefit from Unc5C knockdown to corroborate the results and investigation of the cause-effect relationship. This paper will be of interest to those interested in neural development, sex differences, and/or dopamine function.

---

## [Referee Report · Reviewer #1 (Public Review)]

In this study, Hoops et al. showed that Netrin-1 and UNC5c can guide dopaminergic innervation from nucleus accumbens to cortex during adolescence in rodent models. They found that these dopamine axons project to the prefrontal cortex in a Netrin-1 dependent manner and knocking down Netrin-1 disrupted motor and learning behaviors in mice. Furthermore, the authors used hamsters, a seasonal model that is affected by the length of daylight, to demonstrate that the guidance of dopamine axons is mediated by the environmental factor such as daytime length and in sex dependent manner.

Regarding the cell type specificity of Netrin-1 expression, the authors began by stating "this question is not the focus of the study and we consider it irrelevant to the main issue we are addressing, which is where in the forebrain regions we examined Netrin-1+ cells are present." This statement contradicts the exact issue regarding the specificity issue I raised. They then went on to show the RNAscope data for Netriin-1 in Figure 2, which showed Netrin-1 mRNA was actually expressed quite ubiquitously in anterior cingulate cortex, dorsopeduncular cortex, infralimbic cortex, prelimbic cortex, etc. In addition, contrary to the authors' statement that Netrin-1 is a "secreted protein", the confocal images in Figure 1 in the rebuttal letter actually show Netrin-1 present in "granule-like" organelles inside the cytoplasm of neurons. Finally, the authors presented Figure 7 to indicate the location where virus expressing Netrin-1 shRNA might be located. Again, the brain region targeted was quite focal and most likely did not cover all the Netrin-1+ brain regions in Figure 2. Collectively, these results raised more questions regarding the specificity of Netrin-1 expression in brain regions that are behaviorally relevant to this study.

With respect to the effectiveness of Netrin-1 knockdown in the animals in this study, the authors cited data in HEK293 cells (Figure 5), which did not include any statistics, and previously published in vivo data in a separate, independent study (Figure 6). They do not provide any data regarding the effectiveness of Netrin-1 knockdown in THIS study.

Similar concerns regarding UNC5C knockdown (points #6, #7, and #8) were not adequately addressed.

In brief, while this study provides a potential role of Netrin-1-UNC5C in target innervation of dopaminergic neurons and its behavioral output in risk-taking, the data lack sufficient evidence to firmly establish the cause-effect relationship.

---

## [Referee Report · Reviewer #2 (Public Review)]

In this manuscript, Hoops et al., using two different model systems, identified key developmental changes in Netrin-1 and UNC5C signaling that correspond to behavioral changes and are sensitive to environmental factors that affect the timing of development. They found that Netrin-1 expression is highest in regions of the striatum and cortex where TH+ axons are travelling, and that knocking down Netrin-1 reduces TH+ varicosities in mPFC and reduces impulsive behaviors in a Go-No-Go test. Further, they show that the onset of Unc5 expression is sexually dimorphic in mice, and that in Siberian hamsters, environmental effects on development are also sexually dimorophic. This study addresses an important question using approaches that link molecular, circuit and behavioral changes. Understanding developmental trajectories of adolescence, and how they can be impacted by environmental factors, is an understudied area of neuroscience that is highly relevant to understanding the onset of mental health disorders. I appreciated the inclusion of replication cohorts within the study.

---

## [Referee Report · Reviewer #3 (Public Review)]

This study from the Flores group aims at understanding neuronal circuit changes during adolescence which is an ill-defined, transitional period involving dramatic changes in behavior and anatomy. They focus on DA innervation of the prefrontal cortex, and their interaction with the guidance cue Netrin-1. They propose DA axons in the PFC increase in the postnatal period, and their density is reduced in a Netrin 1 knockdown, suggesting that Netrin abets the development of this mesocortical pathway. In such mice impulsivity gauged by a go-no-go task is reduced. They then provide some evidence that Unc5c is developmentally regulated in DA axons. Finally they use an interesting hamster model, to study the effect of light hours on mesocortical innervation, and make some interesting observations about the timing of innervation and Unc5c expression, and the fact that females housed in winter day length conditions display an accelerated innervation of the prefrontal cortex.

Comments on the revision. Several points were addressed; some remain to be addressed.

#4. It's not clear to me that TH doesn't stain noradrenergic axons in the PFC. See Islam and Blaess, 2021, and references therein.

#6. The Netrin knockdown data provided is from a previous study/samples.

#8. While the authors make the argument that the behavior is linked to DA, they still haven't formally tested it, in my opinion.

#13. Fig 3, UNc 5c levels are not yet quantified. Furthermore, I agree with the previous reviewer that Unc5C knockdown would corroborate key aspects of the model.

New - Developmental trajectory of prefrontal TH-positive axons from early adolescence to adulthood is similar in male and female rats, (Willing Juraska et al., 2017). This needs discussion.

Editors note:

should you choose to revise your manuscript, please include degrees of freedom in your statistical reporting.

---

## [Author Response]

The following is the authors’ response to the previous reviews.

**Public Reviews:**

**Reviewer #1 (Public Review):**
In this study, Hoops et al. showed that Netrin-1 and UNC5c can guide dopaminergic innervation from nucleus accumbens to cortex during adolescence in rodent models.

We showed this with respect to Netrin-1 only. With respect to UNC5c, we showed that the timing of its expression suggests that it may be involved, but did not conduct the UNC5cmanipulation experiments necessary to prove it. We state this clearly in the manuscript.

They found that these dopamine axons project to the prefrontal cortex in a Netrin-1 dependent manner and knocking down Netrin-1 disrupted motor and learning behaviors in mice.

We would like to clarify that we did not show that learning or motor behaviors are affected. We showed that inhibitory control, measured in the Go/No-Go task, is altered in adulthood.

Furthermore, the authors used hamsters, a seasonal model that is affected by the length of daylight, to demonstrate that the guidance of dopamine axons is mediated by the environmental factor such as daytime length and in sex dependent manner.

We agree with this characterization of our hamster experiments, but want to emphasize that it is the *timing* of the adolescent dopamine axon input to the prefrontal cortex what is impacted by daytime length in a sex dependent manner.

Regarding the cell type specificity of Netrin-1 expression, the authors began by stating "this question is not the focus of the study and we consider it irrelevant to the main issue we are addressing, which is where in the forebrain regions we examined Netrin-1+ cells are present." This statement contradicts the exact issue regarding the specificity issue I raised.

We are not sure why the identities of the cell types expressing Netrin-1 are at issue. As a secreted protein, Netrin-1 can be attached to the extracellular cell surface or in the extracellular matrix, where it interacts with its receptors, which are embedded in the cell surfaces of growing axons (Finci et al., 2015; Rajasekharan & Kennedy, 2009). Netrin-1 is expressed by a wide variety of cell types, for example it is expressed in medium spiny neurons in the striatum of rodents as well as in cholinergic neurons (Shatzmiller et al., 2008). However, we cannot see why showing exactly what type(s) of cells have Netrin-1 on their surfaces, or have secreted them into the matrix, would be at issue for our study.

They then went on to show the RNAscope data for Netrin-1 in Figure 2, which showed Netrin-1 mRNA was actually expressed quite ubiquitously in anterior cingulate cortex, dorsopeduncular cortex, infralimbic cortex, prelimbic cortex, etc.

Figure 2 - this is referring to Author response image 2 of our first response to reviewers.

We agree that *Netrin-1* mRNA is present throughout the forebrain. In particular, its presence in the regions mentioned by Reviewer #1 is a key component of our theory for how dopamine axons grow to the prefrontal cortex in adolescence.

In addition, contrary to the authors' statement that Netrin-1 is a "secreted protein", the confocal images in Figure 1 in the rebuttal letter actually show Netrin-1 present in "granule-like" organelles inside the cytoplasm of neurons.

The rebuttal letter’s Figure 1 is not sufficient to determine the subcellular location of the Netrin-1, however we agree that it is likely that Netrin-1 is present in the cytoplasm of neurons. Indeed, its presence in vesicles in the cytoplasm is to be expected as this is a common mechanism for cells to secrete proteins into the extracellular space (Glasgow et al., 2018). We are not sure whether Reviewer #1’s “granule-like” organelles are in fact secretory vesicles or not, and we do not think our immunohistochemical images are an appropriate method by which to determine this kind of question. We find, however, that a detailed characterization of the subcellular distribution of Netrin-1 is beyond the scope of our study.

That Netrin-1 is a secreted protein is well-established in the literature (for example, see Glasgow et al., 2018). The confocal images we provide *suggest,* but do not prove, that it is likely Netrin-1 is present both extracellularly and intracellularly, which is entirely consistent with its synthesis, secretion, and function. It is also consistent with our methodology and findings.

Finally, the authors presented Figure 7 to indicate the location where virus expressing Netrin-1 shRNA might be located. Again, the brain region targeted was quite focal and most likely did not cover all the Netrin-1+ brain regions in Figure 2.

Figure 2 - this is referring to Author response image 2 of our first response to reviewers.

Figure 7 - this is referring to Author response image 4 of our first response to reviewers.

We agree with Reviewer #1’s characterization of our experiment. We intended to interrupt the Netrin-1 pathway to the prefrontal cortex, like removing a bridge along a road. The Netrin-1 signal remained intact along the dopamine axon’s route before and after the location of the viral injection, however it was lost at the site of the virus injection. This is like a road remaining intact on either side of a destroyed bridge, but becoming impassable at the location where the bridge was destroyed. We are glad that Reviewer 1 agrees our experimental design achieved the desired outcome (a focal reduction in Netrin-1 expression).

Collectively, these results raised more questions regarding the specificity of Netrin-1 expression in brain regions that are behaviorally relevant to this study.

We do not agree with this assessment. Our manipulation of Netrin-1 expression was highly localized and specific, as Reviewer #1 seems to acknowledge. We are not clear on what questions this might raise that would call into question our findings as described in our manuscript. We have now added the following paragraph to our manuscript:

“It remains unknown exactly what types of cells are expressing Netrin-1 along the dopamine axon route, and how this expression is regulated to produce the Netrin-1 gradients that guide the dopamine axons. It also remains unclear where the misrouted axons end up in adulthood. Future experiments aimed at addressing these questions will provide further valuable insight into the nature of the “Netrin-1 pathway”. Nonetheless, our results allow us to conclude that Netrin-1 expressing cells “pave the way” for dopamine axons growing to the medial prefrontal cortex.”

With respect to the effectiveness of Netrin-1 knockdown in the animals in this study, the authors cited data in HEK293 cells (Cuesta et al., 2020. Figure 2a), which did not include any statistics, and previously published in vivo data in a separate, independent study (Cuesta et al., 2020. Figure 2c). They do not provide any data regarding the effectiveness of Netrin-1 knockdown in THIS study.

Indeed, we understand the concerns of Reviewer 1 here. This issue was discussed at the time all the experiments (both in the current manuscript and in Cuesta et al., (2020)) were conducted, and we decided that it was sufficient to show the virus was capable of knocking down Netrin-1 in vitro and in vivo in the forebrain. These characterization experiments were published in the first manuscript to present results using the virus, which was Cuesta et al., 2020. However, all experiments from both manuscripts were conducted contemporaneously.

We do not see how repeating the same characterization experiments again is useful.

Similar concerns regarding UNC5C knockdown (points #6, #7, and #8) were not adequately addressed.

There is no UNC5c knockdown in this manuscript. Furthermore, points #6, #7 and #8 do not deal with UNC5c knockdown. Point #6 is regarding the Netrin-1 virus efficacy, which we discuss above. Points #7 and #8 are requesting numerous additional experiments that we feel are worthy of their own manuscripts, and we do not feel that they call into question the findings we present here. Rather, answering points #7 and #8 would further refine our understanding of how dopamine axons grow to the prefrontal cortex beyond our current manuscript.

In brief, while this study provides a potential role of Netrin-1-UNC5C in target innervation of dopaminergic neurons and its behavioral output in risk-taking, the data lack sufficient evidence to firmly establish the cause-effect relationship.

We do not claim a cause-effect relationship here or anywhere in the manuscript. Concrete establishment of a cause-effect relationship will require several more manuscripts worth of experiments.

**Reviewer #2 (Public Review):**
In this manuscript, Hoops et al., using two different model systems, identified key developmental changes in Netrin-1 and UNC5C signaling that correspond to behavioral changes and are sensitive to environmental factors that affect the timing of development. They found that Netrin-1 expression is highest in regions of the striatum and cortex where TH+ axons are travelling, and that knocking down Netrin-1 reduces TH+ varicosities in mPFC and reduces impulsive behaviors in a Go-No-Go test.

We want to point out that we examined the Netrin-1 expression in the septum rather than the striatum but otherwise feel the above description is accurate.

Further, they show that the onset of Unc5 expression is sexually dimorphic in mice, and that in Siberian hamsters, environmental effects on development are also sexually dimorophic. This study addresses an important question using approaches that link molecular, circuit and behavioral changes. Understanding developmental trajectories of adolescence, and how they can be impacted by environmental factors, is an understudied area of neuroscience that is highly relevant to understanding the onset of mental health disorders. I appreciated the inclusion of replication cohorts within the study.

We appreciate Reviewer #2’s comments, which we feel accurately describe our experimental approach and findings, including their limitations.

**Reviewer #3 (Public Review):**
This study from the Flores group aims at understanding neuronal circuit changes during adolescence which is an ill-defined, transitional period involving dramatic changes in behavior and anatomy. They focus on DA innervation of the prefrontal cortex, and their interaction with the guidance cue Netrin1. They propose DA axons in the PFC increase in the postnatal period, and their density is reduced in a Netrin 1 knockdown, suggesting that Netrin abets the development of this mesocortical pathway.

We feel it necessary to point out that we are not the first to propose that dopamine axons in the prefrontal cortex increase in the postnatal period. This is well-established and was first documented in rodents in the 1980s (Kalsbeek et al., 1988). Otherwise we agree with Reviewer 3’s characterization.

In such mice impulsivity gauged by a go-no go task is reduced. They then provide some evidence that Unc5c is developmentally regulated in DA axons. Finally they use an interesting hamster model, to study the effect of light hours on mesocortical innervation, and make some interesting observations about the timing of innervation and Unc5c expression, and the fact that females housed in winter day length conditions display an accelerated innervation of the prefrontal cortex.

We agree with Reviewer #3’s characterization of our study and findings here.

Comments on the revision. Several points were addressed; some remain to be addressed.(4) It's not clear to me that TH doesnt stain noradrenergic axons in the PFC. See Islam and Blaess, 2021, and references therein.

Presuming that Reviewer #3 is referring to Islam et al. (2021), the review they cite supports our position that TH-stained axons in the forebrain are by-and-large dopamine axons.

Nonetheless, Islam et al. do point out that it is important to keep in mind that TH-positive axons have a slight possibility of being noradrenaline axons. We are very conscious of this possibility and are careful to minimize this risk. As we state in the methods, we only examine axons that are morphologically consistent with dopamine axons and are localized to areas within the forebrain where dopamine axons are known to innervate, in addition to being THpositive. The localization and morphology of noradrenaline axons in the forebrain is different from that of dopamine axons. This is stated in our methods on lines 76-94, where we describe in detail the differentiation between dopamine and norepinephrine axons and include a full list of relevant citations.

(6) The Netrin knockdown data provided is from a previous study/samples.

Indeed, however the experiments for the two manuscripts were conducted contemporaneously. We believe two sets of validation experiments are not required.

(8) While the authors make the argument that the behavior is linked to DA, they still haven't formally tested it, in my opinion.

We agree that we have not formally tested this link. However, we disagree that we claim to have established a formal link in our manuscript.

(1). Fig 3, UNc 5c levels are not yet quantified. Furthermore, I agree with the previous reviewer that Unc5C knockdown would corroborate key aspects of the model.

We present UNC5c quantities for mice in our first response to reviewers (Figure 11 therein) however we did not do so for the hamsters due to the time involved. We are planning further experiments with the hamsters and may include quantification of UNC5c in the nucleus accumbens at such time. However, we do not feel its absence from this manuscript calls into question our findings.

With regards to the UNC5c knockdown, we agree it would be an informative extension of our findings here, but again we do not feel that it is necessary to corroborate our current findings.

New - Developmental trajectory of prefrontal TH-positive axons from early adolescence to adulthood is similar in male and female rats, (Willing Juraska et al., 2017). This needs discussion.

Willing et al. (2017) reported an increase in prefrontal dopamine density during adolescence in male and female rats, with a non-significant trend towards an earlier increase in females.

This is in line with our current results in mice indicating that the timing of dopamine axon targeting and growth is sex specific. We are currently testing this idea directly using intersectional viral tracing methods. We now added the following sentence to the manuscript:

“Differences in the precise timing of dopamine innervation to the PFC in adolescence have been suggested by findings reported in male and female rats (Willing et al., 2017)”.

References

Brignani, S., Raj, D. D. A., Schmidt, E. R. E., Düdükcü, Ö., Adolfs, Y., Ruiter, A. A. D., Rybiczka-Tesulov, M., Verhagen, M. G., Meer, C. van der, Broekhoven, M. H., MorenoBravo, J. A., Grossouw, L. M., Dumontier, E., Cloutier, J.-F., Chédotal, A., & Pasterkamp, R. J. (2020). Remotely Produced and Axon-Derived Netrin-1 Instructs GABAergic Neuron Migration and Dopaminergic Substantia Nigra Development. *Neuron*, *107*(4), 684-702.e9. https://doi.org/10.1016/j.neuron.2020.05.037

Cuesta, S., Nouel, D., Reynolds, LM, Morgunova, A., Torres-Berrio, A., White, A., Hernandez, G., Cooper, HM, Flores, C. (2020). Dopamine axon targeting in the nucleus accumbnes in adolescence requires Netrin-1. *Frontiers in Cell and Developmental Biology*, *8*, doi:10.3389/fcell.2020.00487

Finci, L., Zhang, Y., Meijers, R., & Wang, J. H. (2015). Signaling mechanism of the netrin-1 receptor DCC in axon guidance. *Progress in Biophysics and Molecular Biology*, *118*(3), 153-160. https://doi.org/10.1016/j.pbiomolbio.2015.04.001

Glasgow, S. D., Labrecque, S., Beamish, I. V., Aufmkolk, S., Gibon, J., Han, D., Harris, S. N., Dufresne, P., Wiseman, P. W., McKinney, R. A., Séguéla, P., Koninck, P. D., Ruthazer, E. S., & Kennedy, T. E. (2018). Activity-Dependent Netrin-1 Secretion Drives Synaptic Insertion of GluA1-Containing AMPA Receptors in the Hippocampus. *Cell Reports*, *25*(1),

168-182.e6. https://doi.org/10.1016/j.celrep.2018.09.028

Islam, K. U. S., Meli, N., & Blaess, S. (2021). The Development of the Mesoprefrontal Dopaminergic System in Health and Disease. *Frontiers in Neural Circuits*, *15*, 746582. https://doi.org/10.3389/fncir.2021.746582

Kalsbeek, A., Voorn, P., Buijs, R. M., Pool, C. W., & Uylings, H. B. M. (1988). Development of the Dopaminergic Innervation in the Prefrontal Cortex of the Rat. *The Journal of Comparative Neurology*, *269*(1), 58–72. https://doi.org/10.1002/cne.902690105

Rajasekharan, S., & Kennedy, T. E. (2009). The netrin protein family. *Genome Biology*, *10*(9), 239. https://doi.org/10.1186/gb-2009-10-9-239

Shatzmiller, R. A., Goldman, J. S., Simard-Émond, L., Rymar, V., Manitt, C., Sadikot, A. F., & Kennedy, T. E. (2008). Graded expression of netrin-1 by specific neuronal subtypes in the adult mammalian striatum. *Neuroscience*, *157*(3), 621–636. https://doi.org/10.1016/j.neuroscience.2008.09.031

Willing, J., Cortes, L. R., Brodsky, J. M., Kim, T., & Juraska, J. M. (2017). Innervation of the medial prefrontal cortex by tyrosine hydroxylase immunoreactive fibers during adolescence in male and female rats. *Developmental Psychobiology*, *59*(5), 583–589. https://doi.org/10.1002/dev.21525